# Rewired m⁶A epitranscriptomic networks link mutant p53 to neoplastic transformation

An Xu [1,27], Mo Liu[1,27], Mo-Fan Huang [1,2], Yang Zhang [3], Ruifeng Hu[4], Julian A. Gingold [5], Ying Liu[1], Dandan Zhu [1], Chian-Shiu Chien [6,7], Wei-Chen Wang[8], Zian Liao[9], Fei Yuan [9], Chih-Wei Hsu[10], Jian Tu[1], Yao Yu [11], Taylor Rosen [1], Feng Xiong[12], Peilin Jia[4], Yi-Ping Yang[6,7], Danielle A. Bazer[13], Ya-Wen Chen [14,15,16,17], Wenbo Li [2,12], Chad D. Huff[2,11], Jay-Jiguang Zhu [18], Francesca Aguilo [19,20], Shih-Hwa Chiou[6,7], Nathan C. Boles [21], Chien-Chen Lai [8,22,23], Mien-Chie Hung [24,25], Zhongming Zhao [2,4], Eric L. Van Nostrand [9], Ruiying Zhao[1] ✉ & Dung-Fang Lee [1,2,4,26] ✉

N6-methyladenosine (m⁶A), one of the most prevalent mRNA modifications in eukaryotes, plays a critical role in modulating both biological and pathological processes. However, it is unknown whether mutant p53 neomorphic onco-genic functions exploit dysregulation of m⁶A epitranscriptomic networks. Here, we investigate Li-Fraumeni syndrome (LFS)-associated neoplastic transformation driven by mutant p53 in iPSC-derived astrocytes, the cell-of-origin of gliomas. We find that mutant p53 but not wild-type (WT) p53 physically interacts with SVIL to recruit the H3K4me3 methyltransferase MLL1 to activate the expression of m⁶A reader YTHDF2, culminating in an oncogenic phenotype. Aberrant YTHDF2 upregulation markedly hampers expression of multiple m⁶A-marked tumor-suppressing transcripts, including *CDKN2B* and *SPOCK2*, and induces oncogenic reprogramming. Mutant p53 neoplastic behaviors are significantly impaired by genetic depletion of YTHDF2 or by pharmacological inhibition using MLL1 complex inhibitors. Our study reveals how mutant p53 hijacks epigenetic and epitranscriptomic machinery to initiate gliomagenesis and suggests potential treatment strategies for LFS gliomas.

Li-Fraumeni syndrome (LFS) is an autosomal dominant familial cancer syndrome caused by germline *TP53* mutations and characterized by multiple primary neoplasms with early onset[1]. The high tumor incidence in LFS patients provides a strong epidemiological link between p53 mutations and tumorigenesis. Although established LFS mouse models recapitulate certain aspects of the tumor spectrum observed in LFS patients[2,3], mouse models are limited in their ability to dissect the molecular mechanisms of the early stages of LFS neoplastic transformation.

Gliomas rank among the most aggressive and lethal of all human cancers and exhibit a wide range of genetic alterations[4]. While no single genetic alteration defines the disease, the November 2020 iteration of The Cancer Genome Atlas (TCGA) GBM project shows that the p53 pathway (including CDKN2A, MDM2, and p53) is dysregulated in ~85% of tumors, with 28% harboring p53 mutations[5]. The high pre-valence of missense mutations, particularly at certain hotspots, sug-gests that mutant p53 not only provides a selective advantage during cancer progression but also encodes gain-of-function oncoproteins. Indeed, mutant p53 gain-of-function variants have been demonstrated to dysregulate metabolic pathways[6,7], upregulate histone regulators[8], enhance metastasis[9,10] and increase ECM gene expression[11]. Although the above gain-of-function mechanisms have also been proposed to promote mutant p53-associated tumorigenesis, the comprehensive picture of mutant p53's neomorphic oncogenic functions involved in

tumor initiation and progression remains nebulous. Further complicating the picture, most mutant p53 gain-of-function studies have been performed in cancer cell lines characterized by high genome chaos and multiple additional genomic mutations. These broad genomic changes obscure our understanding of the cellular dysfunction directly induced by mutant p53. Therefore, a cellular platform lacking secondary genomic alterations but retaining a strong functional link between p53 mutation and gliomagenesis is required to investigate the oncogenic role of mutant p53 in glioma initiation.

Over 150 types of chemical modifications have been identified as post-transcriptional regulatory marks in multiple RNA species[12]. N6-methyladenosine (m6A) is the most prevalent internal mRNA modification in eukaryotes. m6A is deposited by the m6A methyltransferases METTL3/14 (m6A writer) after coupling with several auxiliary proteins such as WTAP, VIRMA, ZC3H13, HAKAI, and RBM15/15B. m6A marks are removed by the m6A demethylases (erasers) FTO and ALKBH5, and the fates of m6A-modified mRNAs are dependent on the selective m6A readers YTHDF1/2/3, YTHDC1/2, IGF2BP1/2/3, hnRNPA2B1, hnRNPC/G, EIF3, and PRRC2A[13,14]. The functional importance of the m6A modification machinery has been demonstrated in numerous biological processes including mRNA stability and degradation, protein translation, embryonic stem cell self-renewal and differentiation, and DNA damage response[15–20]. Dysregulation of m6A modification and m6A-associated proteins also leads to cancer development[21–23], and dysregulation of the m6A writer METTL3 and m6A reader YTHDC1 modulate mRNA decay to support gliomagenesis[24,25]. However, it is unknown whether genetic alterations (e.g., p53 mutations) cooperatively rewrite m6A epitranscriptomic regulatory networks during glioma development.

In this work, we employ LFS iPSC-derived astrocytes to explore the oncogenic function of mutant p53 in glioma initiation and integrate multilayered regulatory analyses to investigate the epitranscriptomic consequences of mutant p53 gain-of-function. Our studies delineate a dedicated mutant p53 transcriptional complex (mutant p53/SVIL/MLL1) as responsible for decreasing m6A mRNA methylation by increasing YTHDF2 expression. The mutant p53-manipulated epitranscriptomic networks trigger a wide range of downstream effects to drive glioma initiation and tumorigenic cell fate determination.

## Results

### YTHDF2 links mutant p53 to m6A dysregulation in LFS astrocytes

To investigate how mutant p53 triggers glioma initiation, iPSC lines were derived from a LFS patient bearing an inherited germline p53(G245D) mutation with a history of astrocytoma and healthy family controls (wild-type, WT)[26] (Supplementary Fig. 1a). iPSC lines were first differentiated to forebrain-patterned SOX2+/NESTIN+/SOX10− neural progenitor cells (NPCs) and then to GFAP+ astrocytes, the cell-of-origin of astrocytoma, using a modified astrocyte differentiation protocol[27,28] (Fig. 1a and Supplementary Fig. 1b). Differentiated astrocytes at day 75 (D75) presented astrocyte-like morphology and widely expressed GFAP (Fig. 1a). Quantitative RT-PCR (RT-qPCR) confirmed expression of astrocyte-enriched gene ALDH1L1 and astrocyte physiology-related gene SCL1A3 in both WT and LFS iPSC-derived astrocytes (WT and LFS astrocytes) and confirmed relative lack of expression in corresponding NPCs (Supplementary Fig. 1c). These results indicate successful differentiation of WT and LFS iPSCs to the astrocytic lineage.

Because modulations in mRNA m6A levels have been recognized to impact multiple pathological features of GBM (WHO grade IV astrocytoma) and glioblastoma stem cells (GSCs)[29,30], we sought to determine if dysregulation of global m6A modifications could be identified in LFS astrocytes. Strikingly, LFS astrocytes displayed consistently decreased global m6A levels compared to WT astrocytes (Fig. 1b and Supplementary Fig. 2a). Moreover, depletion of p53 and mutant p53 by two independent, non-overlapping small hairpin RNAs

(shRNAs) consistently led to elevated global m6A level in LFS astrocytes (Fig. 1c and Supplementary Fig. 2b); in contrast, knockdown of p53 did not alter the global m6A profile in WT astrocytes (Supplementary Fig. 2b, c), suggesting mutant p53-dependent regulation of m6A modifications. Taken together, our data reveal that regulation of m6A machinery in astrocytes by mutant p53 but not WT p53 contributes to globally decreased m6A mRNAs.

To elucidate the neomorphic function of mutant p53 in regulating the m6A-associated epitranscriptome, we explored the global transcriptomes of WT and LFS astrocytes by RNA-seq. Among 20 known m6A regulators, we found that YTHDF2, but not other m6A regulators, was significantly upregulated in LFS astrocytes (Fig. 1d). The upregulation of YTHDF2 in astrocytes derived from both LFS iPSCs and a TALEN engineered heterozygous p53(G245D) H1 hESC line (H1-p53(WT/G245D)) was further validated by immunoblotting (Fig. 1e). Immunoblotting further validated the lack of difference of protein expression of METTL3, METTL14, WTAP, FTO, and ALKBH5 between WT and LFS astrocytes (Supplementary Fig. 2d). YTHDF2 functions as an m6A reader to drive m6A-modified mRNA toward degradation[31]. Knockdown of YTHDF2 led to elevated m6A levels in WT and LFS astrocytes (Supplementary Fig. 2e, f), confirming its function in the astrocytic context. To determine if aberrant YTHDF2 upregulation depends on mutant p53, we depleted p53 and mutant p53 by shRNAs and examined YTHDF2 expression. Knockdown of mutant p53 led to significant downregulation of YTHDF2 mRNA and protein expression in LFS astrocytes; in contrast, depletion of p53 resulted in limited effects on YTHDF2 expression in WT astrocytes (Fig. 1f, g), suggesting that mutant p53 distinctly modulates YTHDF2 transcription. To exclude the possibility of mutant p53 regulating YTHDF2 expression only in iPSC and hESC platforms, we stably transduced mutant p53 into a p53-null GBM cell line LNZ308. Ectopic expression of mutant p53 in LNZ308 cells consistently increased both YTHDF2 mRNA and protein (Supplementary Fig. 2g). We next examined if YTHDF2 expression can be regulated by other mutant p53s. Consistent with our p53(G245D)) data, numerous hotspot mutant p53s, including p53(R175H), p53(R248W), p53(R249S), p53(R273H), and p53(R280T), also markedly induced YTHDF2 expression in astrocytes (Fig. 1h, i and Supplementary Fig. 2h). These findings suggest that mutant p53-regulated YTHDF2 expression is a general phenomenon among astrocytes and astrocyte-derived malignancies and that transcriptional regulation of YTHDF2 is a shared function among distinct mutant p53s.

### YTHDF2 is required for mutant p53-induced oncogenic transformation

In light of emerging evidence implicating m6A mRNA modifications in gliomagenesis[29,30], we examined whether mutant p53-regulated YTHDF2 expression contributes to LFS astrocyte-associated cell proliferation and neoplastic transformation, an oncogenic event that occurs during the early stages of tumor initiation. Cerebral organoid culture demonstrated markedly increased LFS organoid size, indicating that LFS differentiated cells are more proliferative than WT counterparts in a 3D cerebral developmental environment (Fig. 2a and Supplementary Fig. 3a); in contrast, depletion of YTHDF2 significantly impaired the growth of LFS cerebral organoids (Fig. 2a). Furthermore, LFS astrocytes maintained a high proliferation rate after 4 months of culture (Supplementary Fig. 3b). In vitro anchorage-independent growth (AIG) assay demonstrated that LFS astrocytes but not WT astrocytes undergo neoplastic transformation and that LFS astrocytes exhibit a steady increase in growth and doubling in number between D30 and D60 (Supplementary Fig. 3c). Depletion of YTHDF2 hampered colony growth of LFS astrocytes in soft agar (Fig. 2b), suggesting that YTHDF2 plays an essential role in mutant p53-mediated oncogenic features in LFS cells. Collectively, these results indicate that LFS astrocytes acquire early advantages in cell proliferation and oncogenic transformation that may contribute toward glioma initiation.

Complete YTHDF2 depletion has been demonstrated to lead to differentiation failure and unexpected cell death[32]. To investigate if YTHDF2 is required for in vivo engraftment of LFS cerebral organoids, we partially depleted YTHDF2 mRNA in LFS iPSCs to the level of WT iPSCs by adjusting the titers of lentiviral shYTHDF2. As expected, following 1 month in 3D culture, cerebral organoids formed and demonstrated abundant PAX6⁺/SOX2⁺ forebrain cell populations (Supplementary Fig. 3d, left panel). RT-qPCR verified

lower *YTHDF2* mRNA expression in LFS/shYTHDF2 cerebral organoids compared with LFS/shCtrl cerebral organoids and comparable *YTHDF2* mRNA expression to WT/shCtrl cerebral organoids (Supplementary Fig. 3d, right panel). Dissected cerebral organoid pieces of 1.5 mm diameter from each of the organoid lineages were transplanted into immunodeficient mouse cortex using an established brain organoid transplantation protocol[33,34] and cerebral organoid engraftments were examined 2 months following

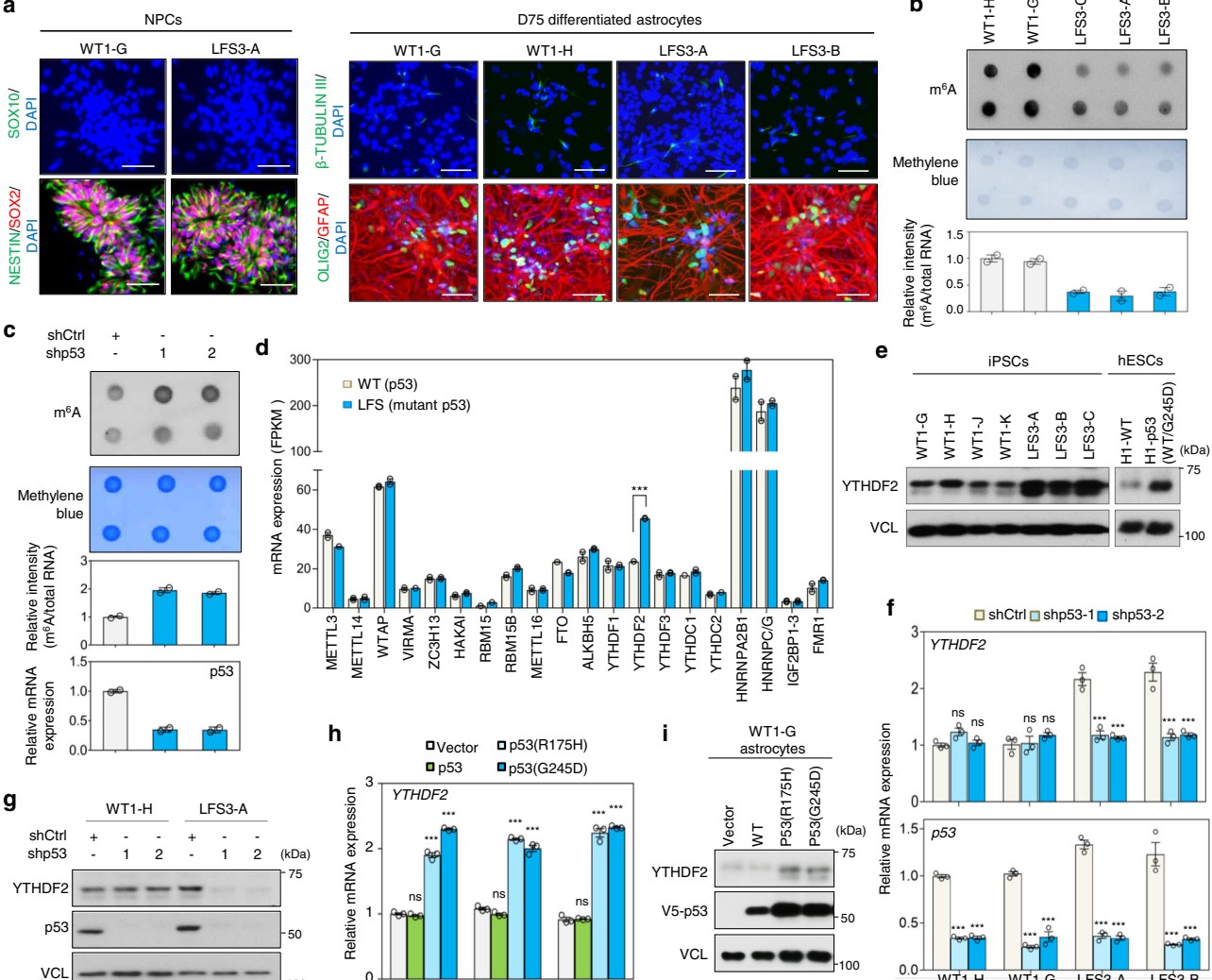

**Fig. 1 | Mutant p53 upregulates m⁶A reader YTHDF2 expression in LFS astrocytes and glioma cells. a** Immunostaining indicates iPSC-derived NPCs and astrocytes expressing their corresponding cell markers (SOX2 and NESTIN for NPCs, GFAP for astrocytes, OLIG2 for oligodendrocytes, and β-TUBULIN III for neurons). Scale bar, 50 μm. **b** m⁶A methylation dot blotting shows decreased m⁶A methylation in LFS astrocytes. Dot blotting is performed to identify polyadenylated mRNAs immunoblotted with anti-m⁶A antibodies (upper panel). Methylene blue staining of total mRNA is used as a loading control (lower panel). Dot density is measured by ImageJ. The blotting images represent the results of at least three independent experiments, while the bar charts depict technical replicates within a single experiment. **c** m⁶A methylation dot blotting indicates increased m⁶A methylation upon depletion of mutant p53 in LFS astrocytes. Dot blotting identifies polyadenylated mRNA isolated from shCtrl and shp53 transduced LFS astrocytes and immunoblotted with anti-m⁶A antibodies (upper panel). Methylene blue staining of total mRNA is used as a loading control (lower panel). Dot density is measured by ImageJ. The blotting images represent the results of at least three independent experiments, while the bar charts depict technical replicates within a single experiment. **d** Transcriptome analysis of the mRNA expression of known m⁶A

regulators in WT and LFS astrocytes. Among 20 m⁶A regulators examined in this study, m⁶A reader YTHDF2 is significantly upregulated in LFS astrocytes compared with WT astrocytes (n = 2 biologically independent samples). **e** Immunoblotting indicates elevated YTHDF2 protein in multiple LFS and H1-p53(WT/G245D) astrocytes compared with WT and H1-WT astrocytes. **f** RT-qPCR analysis shows a decrease of YTHDF2 expression upon p53/mutant p53 knockdown in LFS astrocytes but not WT astrocytes (n = 3 biologically independent samples). **g** Depletion of p53/mutant p53 by p53 shRNAs leads to downregulated YTHDF2 protein expression in LFS astrocytes but not WT astrocytes. **h** RT-qPCR shows mutant p53s (p53(R175H) and p53(G245D)) upregulate *YTHDF2* mRNA expression in WT astrocytes (n = 3 biologically independent samples). **i** Immunoblotting indicates upregulation of YTHDF2 protein following transduction of distinct mutant p53s (p53(R175H) and p53(G245D)) but not p53 into WT iPSC-derived astrocytes. The results are representative of at least three independent experiments (**a–c**, **e**, **g**, **i**). The data are presented as the mean ± SEM; two-way ANOVA with Bonferroni's multiple comparison test (**h**, **f**); multiple *t* test (**d**). ***P < 0.001. ns not significant. Source data and exact *P* values are provided in the Source Data file.

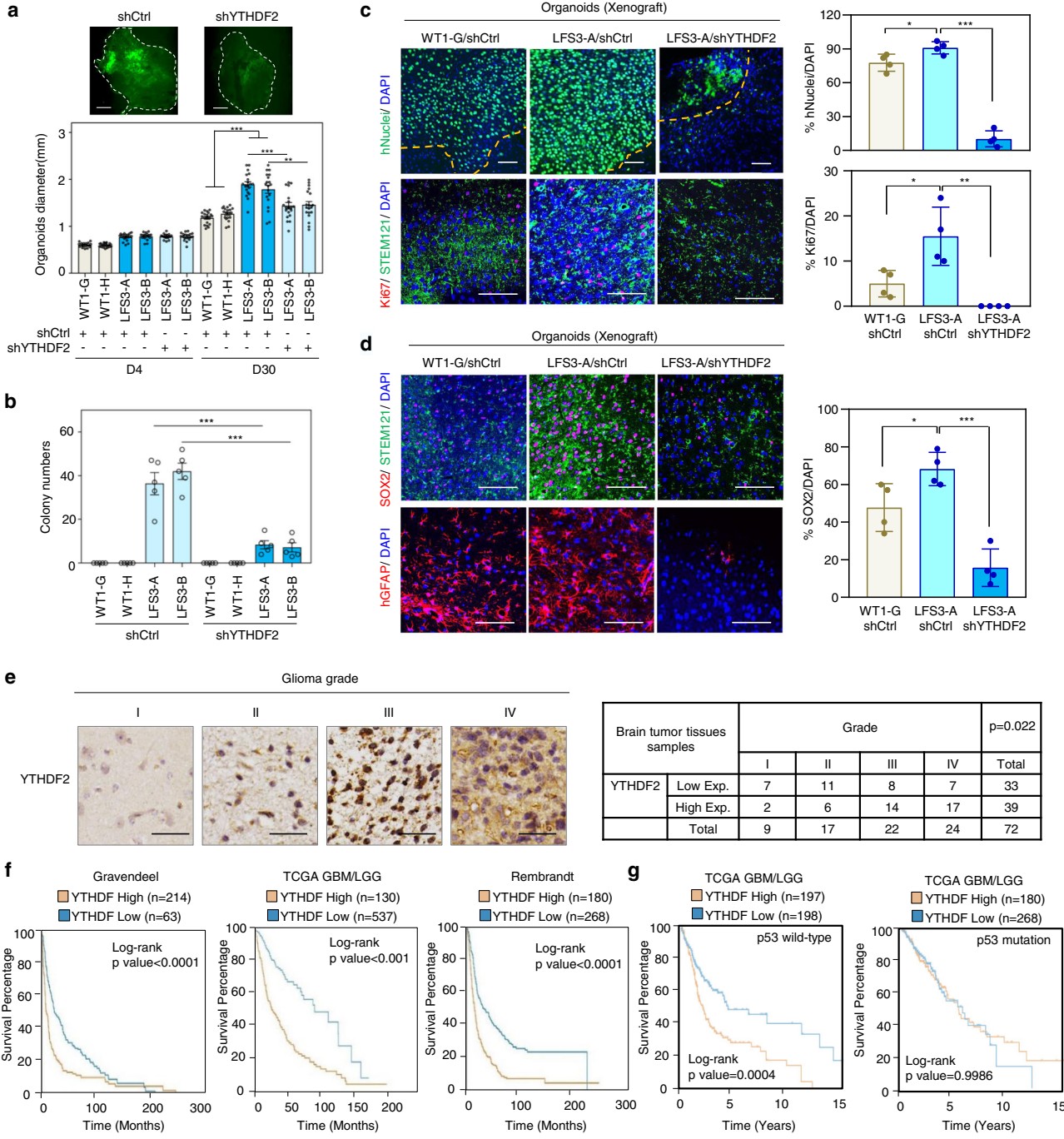

**Fig. 2 | YTHDF2 is associated with mutant p53-induced neomorphic oncogenic function and its expression is correlated with poor prognosis in glioma patients. a** Knockdown of YTHDF2 leads to decreased LFS cerebral organoid development. The average diameters of shCtrl- and shYTHDF2-transduced LFS organoids and WT organoids are quantified at 4 and 30 days (*n* = 19 biologically independent samples). Scale bar, 500 μm. **b** In vitro AIG assay demonstrates decreased colony numbers upon YTHDF2 depletion in LFS astrocytes (*n* = 5 biologically independent samples). All colonies are counted and measured after 2-month culture. **c** Immunofluorescence staining of engrafted cerebral organoids in mouse cortices is used to determine organoid size and cell proliferation. Upper panel: Organoid engraftment is determined by the presence of human nuclear antigen (hNuclei) in mouse cortex and quantified by the percentage of stained (mouse or human) nuclei (DAPI) in the microscopic field staining for hNuclei. Lower panel: The proliferative human cells (STEM121⁺ cells) in engrafted organoids are determined by quantifying the percentage of Ki67 over DAPI (*n* = 4 biologically independent samples). Bar plots display the hNuclei⁺/DAPI⁺ and Ki67⁺/DAPI⁺ ratios.

The boundary between organoids and mouse brain is shown as a dashed line. Scale bar, 100 μm. **d** SOX2 immunofluorescence staining indicates LFS cerebral organoids maintain progenitor characteristics in vivo (*n* = 4 biologically independent samples). Bar chart indicates the SOX2/DAPI ratio for each experimental condition. Scale bar, 100 μm. **e** IHC studies indicate elevated YTHDF2 expression in higher-grade gliomas. YTHDF2 expression is analyzed by IHC (*n* = 72 primary human glioma specimens). Representative specimens of different glioma grades are shown in the top panels. Scale bar, 25 μm. **f** Multiple glioma datasets suggest that high YTHDF2 expression is correlated with poor overall survival of glioma patients. Log-rank (Mantel–Cox) test is performed to compute significance. **g** Overall survival in p53 WT and mutant glioma patients with high or low YTHDF2 gene expression in TCGA GBM/LGG dataset. The data are presented as the mean ± SEM; one-way ANOVA with Tukey's multiple comparison test (**b**); two-way ANOVA with Bonferroni's multiple comparison test (**a**); unpaired two-tailed Student's *t* test (**c**, **d**); *P* < 0.05, **P* < 0.01, ***P* < 0.001. Source data and exact *P* values are provided in the Source Data file.

transplantation (Supplementary Fig. 3e). Both WT and LFS cerebral organoids exhibited robust integration and hyperproliferation in mouse cortex and were distributed throughout most of the implantation cavity, as determined by the abundant expression of human nuclear antigens (hNuclei) among all (DAPI-staining) nuclei and verified by the high hNuclei/DAPI ratio. The hNuclei/DAPI ratio revealed that LFS cerebral organoids demonstrated more effective engraftment compared with WT counterparts, while knockdown of YTHDF2 markedly hampered the engraftment of LFS cerebral organoids (Fig. 2c, upper panels). LFS grafts demonstrated the highest populations of Ki67+ proliferative cells among the three groups of grafts, whereas few Ki67+ proliferative cells were observed in LFS/shYTHDF2 grafts (Fig. 2c, lower panels), indicating the essential role of YTHDF2 in LFS cell hyperproliferation in vivo. Interestingly, although WT and LFS astrocytes did not differ in SOX2 expression in 2D monolayer cultured NPCs (Fig. 1a), LFS grafts exhibited significantly more SOX2 expression than WT grafts and SOX2 expression in LFS grafts was abrogated by YTHDF2 knockdown (Fig. 2d), suggesting that mutant p53 may contribute to LFS-associated tumor initiation via formation of a poorly differentiated SOX2+ cell population. Furthermore, consistent with the essentiality of Ythdf2 in mouse glia development[32], depletion of YTHDF2 resulted in a decreased proportion of GFAP + cells in LFS organoids (Fig. 2d, lower panels).

To determine the clinical relevance of YTHDF2 in gliomagenesis, we examined YTHDF2 expression across 72 human glioma specimens by immunohistochemical (IHC) staining. Increased YTHDF2 expression by IHC was associated with higher-grade gliomas (Fig. 2e). We next examined the expression of YTHDF2 across three distinct public brain tumor datasets (Gravendeel, TCGA, and Rembrandt). Consistent with the IHC observations, YTHDF2 levels were higher in tumors compared with normal tissues, and higher YTHDF2 expression was associated with higher tumor grades (Supplementary Fig. 3f, g). High expression of YTHDF2 among patients with gliomas was also associated with poor survival (Fig. 2f). Interestingly, high expression of YTHDF2 was associated with poor survival in glioma patients with WT p53 but not mutant p53 (Fig. 2g), suggesting that high YTHDF2 expression is correlated to poor prognosis in p53 WT glioma patients and the elevated level of YTHDF2 by mutant p53 is a sufficient dominant influencer of patient survival among p53 mutant glioma patients. Collectively, these findings demonstrate that mutant p53-induced YTHDF2 plays an oncogenic role in glioma development and that elevated YTHDF2 expression is associated with poor clinical outcomes among glioma patients.

### Genome-wide ChIP-seq mapping reveals both p53 and mutant p53 binding to the *YTHDF2* promoter

To explore how mutant p53 transcriptionally regulates YTHDF2 expression and thereby drives oncogenic programming in LFS astrocytes, we carried out p53 chromatin immunoprecipitation followed by next-generation sequencing (ChIP-seq) to determine global genome occupancies by p53 and mutant p53 in WT and LFS astrocytes, respectively (Supplementary Data 1). The binding pattern of p53/mutant p53 to peak-proximal regions (within 3 kb) was analyzed. In agreement with the previous reports[8,35,36], the genome-wide p53 and mutant p53 binding regions showed relatively little overlap, with 2,094 shared peaks, 11,147 p53-unique peaks, and 1137 mutant p53-unique peaks identified among WT and LFS astrocytes (Fig. 3a and Supplementary Fig. 4a). Compared to the p53 genome-binding loci in WT astrocytes, the mutant p53's genome-binding loci in LFS astrocytes primarily mapped to promoters and gene coding regions (Supplementary Fig. 4b). Motif analysis of p53/mutant p53 peaks identified enrichment of canonical p53 and p63 binding motifs in both p53-unique and shared peak groups; in contrast, RREB1 and KLF9 binding motifs were slightly enriched among the

mutant p53-unique peaks (Supplementary Fig. 4c). These results indicate that mutant p53 retains certain p53 functions while gaining additional genome-binding abilities to modulate new categories of gene expression in LFS astrocytes. Interestingly, analyses of the ChIP-seq read distribution illustrated that both p53 and mutant p53 bind equally to the H3K27Ac-marked *YTHDF2* promoter in WT and LFS astrocytes but that mutant p53 has significantly reduced binding to the *p21* promoter (Fig. 3b). ChIP-qPCR further validated p53 and mutant p53 binding to the *YTHDF2* promoter and lack of binding to regions upstream of the *YTHDF2* promoter peak region in multiple WT and LFS astrocytes (Fig. 3c). To determine whether WT p53 is required for mutant p53s to bind to the *YTHDF2* promoter, we transfected WT and mutant p53s into LNZ308 (p53-null) glioma cells. Examination of chromatin binding by ChIP-qPCR revealed that both WT and mutant p53s are capable of directly binding to the *YTHDF2* promoter (Fig. 3d). Electrophoretic mobility shift assay (EMSA) further demonstrated that mutant p53 directly binds to the *YTHDF2* promoter and forms a complex in vitro (Fig. 3e). Together, these studies demonstrate that YTHDF2 is a direct mutant p53 transcriptional target.

### Mutant p53 interactome links SVIL to YTHDF2 transcription

Although our ChIP-seq results implied that YTHDF2 can be a transcriptional target of both p53 and mutant p53, only mutant p53 regulated YTHDF2 expression (Fig. 1f–i), suggesting the involvement of an additional regulatory layer. Hence, we hypothesized that an unidentified mutant p53-interacting molecule mediated mutant p53-driven YTHDF2 expression. Immunoprecipitation and mass spectrometry (IP-MS) identified 139 proteins associated with both p53 and mutant p53 in WT and LFS astrocytes, 89 proteins specifically bound to mutant p53 in LFS astrocytes and 162 proteins specifically bound to p53 in WT astrocytes with high confidence (Fig. 3f and Supplementary Data 2). Since histone modifications are widely recognized to modulate gene expression, we chose to further investigate the histone methyltransferase binding protein, SVIL, among the proteins identified to selectively bind mutant p53 (Supplementary Fig. 4d). Co-immunoprecipitation (co-IP) by pulling down V5-tagged p53 or mutant p53 demonstrated that mutant p53 but not p53 physically interacts with SVIL (Fig. 3g). This was confirmed by reciprocal co-IP by immunoprecipitating GFP-tagged SVIL (Supplementary Fig. 4e). The physical interaction between mutant p53 and SVIL was also validated with endogenous co-IP (Fig. 3h). To further explore if the mutant p53 and SVIL interaction also exists in other hotspot missense mutant p53s, we investigated the physical interaction between p53(R175H) and SVIL and confirmed a strong binding affinity (Supplementary Fig. 4f), suggesting that the physical interaction of SVIL with distinct mutant p53s is a general phenomenon. To determine if the mutant p53 and SVIL interaction is retained under stress conditions activating p53, we treated LFS astrocytes with etoposide and again confirmed the persistence of an interaction between mutant p53 and SVIL (Fig. 3i), suggesting that p53 activity does not affect the specificity of mutant p53 association with SVIL. Depletion of mutant p53 impaired SVIL binding on the *YTHDF2* promoter, emphasizing that mutant p53 interacts with and recruits SVIL to the *YTHDF2* promoter (Fig. 3j). We mapped the specific mutant p53 domain that interacts with SVIL using a series of domain constructs. The DNA binding domain (DBD) of mutant p53s, including the DBDs of p53(R175H) and p53(G245D) is essential for mutant p53 and SVIL interaction (Supplementary Fig. 4g), indicating that mutant p53 interacts with SVIL through a mutated p53 DBD.

Knockdown of SVIL led to YTHDF2 downregulation in LFS astrocytes and in GBM cells expressing mutant p53 (LNZ308-p53(G245D)) but not in WT astrocytes and in LNZ308-Vector cells (Fig. 3k and Supplementary Fig. 4h, i). Depletion of both mutant p53 and SVIL did

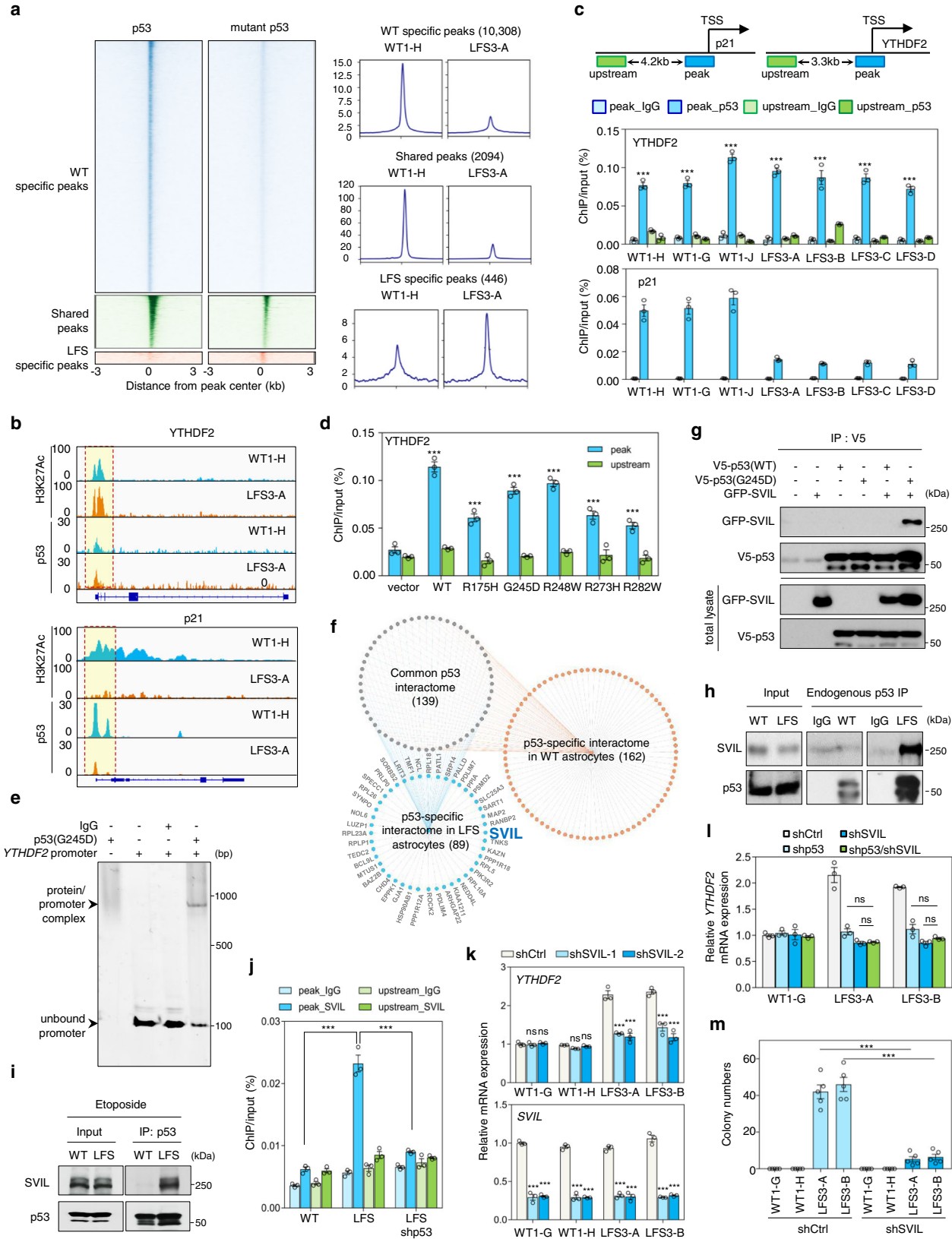

not lead to additional downregulation of YTHDF2 expression compared to knockdown of mutant p53 or SVIL alone (Fig. 3l), indicating that SVIL regulation of YTHDF2 expression is dependent on mutant p53. AIG and fluorescence-based competition assay demonstrated that knockdown of SVIL results in the impairment of neoplastic transformation and cell proliferation of LFS astrocytes, respectively (Fig. 3m

and Supplementary Fig. 4j). Furthermore, clonogenic assay demonstrated that suppression of SVIL dramatically decreases LNZ308-p53(G245D) cell survival compared to LNZ308-Vector cells (Supplementary Fig. 4k). Taken together, these results suggest that SVIL is functionally involved in mutant p53-mediated YTHDF2 transcription through direct interaction with mutant p53.

**Fig. 3 | Genome occupancy and interactome studies reveal that mutant p53 cooperates with SVIL in regulating YTHDF2 expression. a** Heatmaps (left panels) depict the p53/mutant p53 genomic occupancies of p53-specific, mutant p53-specific, and shared peaks within 3 kb of peak centers according to p53 ChIP-seq. Composite plots (right panels) show normalized p53 and mutant p53 density distribution at promoters of genes within p53-specific, mutant p53-specific, and shared peaks. **b** Integrative Genomics Viewer (IGV) track views of H3K27ac, p53, and mutant p53 genome occupancy over *p21* and *YTHDF2* promoter regions in multiple WT and LFS astrocytes. **c** ChIP-qPCR validation of p53 and p53(G245D) binding peaks at the identified *YTHDF2* promoter. p53/mutant p53 binding peak (blue) and upstream non-p53/mutant p53 binding regions (green) used for ChIP-qPCR validation (*n* = 3 biologically independent samples). **d** ChIP-qPCR indicates that p53 and various p53 mutants bind to the *YTHDF2* promoter but not the upstream non-p53/mutant p53 binding region (*n* = 3 biologically independent samples). **e** EMSA demonstrates direct p53(G245D) binding to the *YTHDF2* promoter. **f** Snapshot of p53 and mutant p53 interaction network. Connectivity map between p53 and mutant p53 interactomes in WT and LFS astrocytes. **g** p53(G245D) but not p53 interacts with SVIL exogenously. **h** Endogenous interaction between p53(G245D) and SVIL in LFS astrocytes. **i** Etoposide-induced WT p53 activation does not alter the interaction of p53(G245D) and SVIL in LFS astrocytes. **j** ChIP-qPCR indicates that knockdown of p53(G245D) hampers SVIL binding on the *YTHDF2* promoter in LFS astrocytes (*n* = 3 biologically independent samples). **k** RT-qPCR analysis shows decreased *YTHDF2* mRNA expression upon SVIL knockdown in LFS astrocytes but not WT astrocytes (*n* = 3 biologically independent samples). **l** RT-qPCR analysis demonstrates comparable YTHDF2 mRNA expression upon depletion of mutant p53, SVIL, and mutant p53/SVIL in LFS astrocytes (*n* = 3 biologically independent samples). **m** In vitro AIG assay demonstrates decreased colony numbers upon SVIL depletion in LFS astrocytes (*n* = 5 biologically independent samples). The results are representative of at least three independent experiments (**e**, **g**–**i**). The data are presented as the mean ± SEM; two-way ANOVA with Bonferroni's multiple comparison test (**c**, **d**, **j**–**m**). ***$P < 0.001$. ns not significant. Source data and exact $P$ values are provided in the Source Data file.

## Mutant p53/SVIL recruitment of MLL1 activates YTHDF2 expression and promotes oncogenic features

Analyzing transcriptome data of mutant p53 astrocytes, we noticed an enrichment of H3K4me3-regulated genes in both LFS and H1-p53(WT/G245D) astrocytes (Fig. 4a). IP-MS analysis of SVIL-associated proteins identified 239 high-confidence SVIL-interacting proteins, including the H3K4me3 methyltransferase MLL1, also known as lysine methyltransferase 2A (KMT2A) (Fig. 4b and Supplementary Data 3). Co-IP assay verified that endogenous MLL1 physically interacts with SVIL in GFP-SVIL transduced LFS astrocytes (Fig. 4c). Proximity ligation assay (PLA) validated that endogenous mutant p53 forms a complex with SVIL and MLL1 in LFS astrocytes (Fig. 4d). Depletion of SVIL hampered p53(G245D)/SVIL/MLL1 complex formation (Fig. 4e, f) and knockdown of MLL1 did not impair mutant p53 and SVIL interaction (Fig. 4f), indicating that SVIL functions as a bridge to recruit MLL1 to mutant p53-bound promoter regions to activate YTHDF2 transcription via increasing H3K4me3. Furthermore, H3K4me3 ChIP-seq analysis revealed more prominent H3K4me3 peaks on the *YTHDF2* promoter in LFS astrocytes compared with WT astrocytes (Fig. 4g). Pharmacological inhibition of MLL1 by OICR-9429 decreased H3K4me3 peaks on the *YTHDF2* promoter in LFS astrocytes but had only limited effects on H3K4me3 in WT astrocytes (Fig. 4g), suggesting that MLL1-regulated H3K4me3 on the *YTHDF2* promoter is mutant p53-dependent.

Knockdown of MLL1 decreased YTHDF2 expression in LFS but not in WT astrocytes (Fig. 4h), and MLL1 inhibition with OICR-9429 and MI-2-2 in LFS astrocytes markedly reduced YTHDF2 protein expression (Fig. 4i), suggesting the essential function of MLL1 in mutant p53-mediated YTHDF2 expression. OICR-9429 and MI-2-2 selectively reduced cell viability of LFS astrocytes in both dose- and time-dependent manners compared with WT astrocytes (Fig. 4j). Similarly to YTHDF2 and SVIL, depletion of MLL1 impaired the in vitro oncogenic transformation ability of LFS astrocytes (Fig. 4k). The decreased cell proliferation and colony numbers upon SVIL or MLL1 knockdown were rescued by YTHDF2 overexpression in LFS astrocytes (Fig. 4l, m), suggesting that YTHDF2 is downstream of SVIL and MLL1 in LFS cell viability and oncogenic transformation. Consistently, OICR-9429 and MI-2-2 selectively suppressed the clonogenic ability of LNZ308-p53(G245D) compared with cells from the LNZ308-Vector GBM line (Fig. 4n).

We next investigated if MLL1 inhibition can also selectively hamper LFS cell growth in a 3D cerebral developmental environment. mCherry⁺ WT iPSCs and GFP⁺ LFS iPSCs were mixed at a 1:1 ratio and differentiated in a 3D co-culture system to form cerebral organoids (Supplementary Fig. 5a). After 2 months of culture, nineteen organoids were treated with MLL1 inhibitor OICR-9429 and followed for 7 days, with the ratios of mCherry⁺ WT and GFP⁺ LFS cell populations examined by light sheet confocal imaging and flow cytometry. Light sheet imaging showed a dramatic decrease in the GFP/mCherry ratio upon

OICR-9429 treatment (Fig. 4o and Supplementary Fig. 5b), which was validated by flow cytometry analysis (Fig. 4p), indicating that LFS cells are more sensitive to MLL1 inhibitors than WT cells in a 3D cerebral developmental environment.

## Integration of m⁶A MeRIP-seq, eCLIP-seq, and RNA-seq identifies YTHDF2 targets in LFS astrocytes

YTHDF2-mediated m⁶A mRNA degradation is critical for premalignant cells to downregulate specific tumor suppressor transcripts to influence tumorigenesis[29,37,38], but it is unclear if this epitranscriptomic regulation is involved in mutant p53-mediated gliomagenesis. To dissect which mRNA transcripts are direct targets of YTHDF2 in LFS astrocytes, we combined m⁶A MeRIP-seq to determine the transcriptome-wide distribution of m⁶A mRNAs with enhanced cross-linking and immunoprecipitation (eCLIP)[39] to identify YTHDF2-specific targets. Analysis of the m⁶A peak distribution in LFS astrocytes revealed that m⁶A is predominantly found on 5'UTRs and 3'UTRs of actively transcribed mRNAs, as expected (Fig. 5a and Supplementary Data 4). In comparison with transcripts with no or low m⁶A modification, transcripts with high m⁶A modification were significantly elevated upon YTHDF2 depletion in LFS astrocytes (Fig. 5b), supporting the negative regulation of m⁶A-modified transcripts by YTHDF2. YTHDF2 eCLIP was then performed in two independent experiments (Fig. 5c). In agreement with a previous YTHDF2 eCLIP study[40], the YTHDF2-interacting mRNA peaks mapped to 3'UTRs clustered around stop codons (Fig. 5d, Supplementary Fig, 6a, and Supplementary Data 5). Motif analysis established that YTHDF2 binding motifs match the consensus m⁶A motif RRACH (R represents A or G, A represents m⁶A, and H represents A, C, or U) in LFS astrocytes, with GGACU being the most common consensus sequence (Fig. 5e and Supplementary Fig. 6b).

By integrating m⁶A MeRIP-seq, YTHDF2 eCLIP-seq, and RNA-seq studies, we narrowed our attention from the 1,352 transcripts upregulated upon YTHDF2 depletion to those also with m⁶A-markings and downregulated in LFS astrocytes compared with WT astrocytes (Fig. 5f). This subset would capture transcripts directly targeted by YTHDF2 for degradation via m⁶A as a consequence of p53 mutation. Among the 84 transcripts identified as directly targeted by YTHDF2 in LFS astrocytes, we examined 2 representative genes *CDKN2B* and *SPOCK2* (Fig. 5f). We confirmed the existence of m⁶A and YTHDF2 binding peaks in *CDKN2B* and *SPOCK2* transcripts (Fig. 5g) and decreased *CDKN2B* and *SPOCK2* expression in LFS compared with WT astrocytes (Fig. 5h). Low expression of CDKN2B and SPOCK2 was associated with poor survival in glioma patients in the TCGA LGG/GBM dataset (Fig. 5i). Consistently, higher YTHDF2 expression and lower CDKN2B and SPOCK2 expression were observed in LFS cerebral organoids engrafted in the mouse cortex (Fig. 5j and Supplementary Fig. 6c, d). Depletion of YTHDF2 led to upregulation of

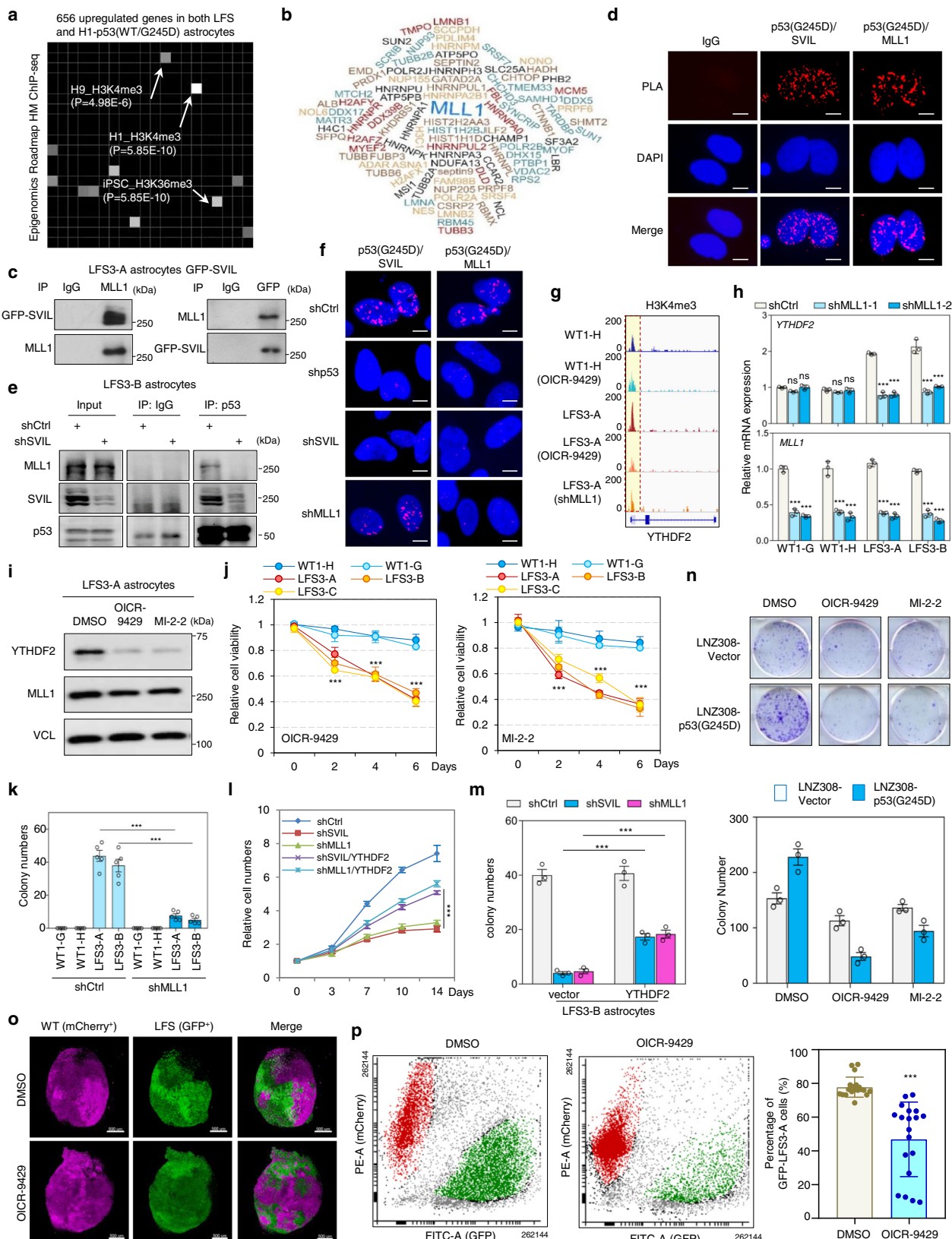

CDKN2B and SPOCK2 in engrafted LFS organoids (Fig. 5j and Supplementary Fig. 6c, d), Depletion of YTHDF2 led to an increase in the half-life of *CDKN2B* and *SPOCK2* mRNAs (Fig. 5k), indicating that YTHDF2 directly regulates *CDKN2B* and *SPOCK2* mRNA degradation. m⁶A MeRIP-PCR indicated that depletion of YTHDF2 leads to an increase of m⁶A markings on *CDKN2B* and *SPOCK2* transcripts

(Supplementary Fig. 6e). Depletion of mutant p53, YTHDF2, and SVIL, as well as inhibition of MLL1 function by OICR-9429 and MI-2-2, led to increased expression of *CDKN2B* and *SPOCK2* mRNAs in LFS astrocytes (Fig. 5l). Collectively, these findings establish the pathological role of mutant p53/SVIL/MLL1 in modulating YTHDF2-mediated m⁶A-marked mRNA degradation.

**Fig. 4 | MLL1 is recruited by mutant p53/SVIL to activate YTHDF2 expression.**
**a** Epigenomics Roadmap indicates that genes with H3K4me3 peaks in their pro-
moter regions are enriched in LFS astrocytes. **b** Word clouds represent proteins
inferred with high confidence to interact with SVIL in LFS astrocytes. **c** Endogenous
interaction between SVIL and MLL1 in LFS-GFP-SVIL astrocytes. **d** PLA analysis
indicates that endogenous mutant p53 forms a complex with SVIL and MLL1 in LFS
astrocytes. Scale bar, 10 μm. **e** Depletion of SVIL impairs p53(G245D)/SVIL/MLL1
complex formation in LFS astrocytes. **f** Depletion of SVIL impairs mutant p53/SVIL/
MLL1 complex formation. Scale bar, 10 μm. **g** H3K4me3 peaks on the *YTHDF2*
promoter are reduced upon MLL1 depletion or MLL1 inhibition by OICR-9429 in LFS
astrocytes. **h** RT-qPCR analysis demonstrates decreased *YTHDF2* mRNA expression
upon MLL1 knockdown in LFS astrocytes but not WT astrocytes (n = 3 biologically
independent samples). **i** Immunoblotting indicates reduced YTHDF2 protein upon
the treatment with MLL1 inhibitors OICR-9429 and MI-2-2 in LFS astrocytes. **j** OICR-
9429 and MI-2-2 selectively inhibit cell proliferation of LFS astrocytes (n = 5 biolo-
gically independent samples). **k** In vitro AIG assay demonstrates decreased colony

numbers upon MLL1 depletion in LFS astrocytes (n = 5 biologically independent
samples). **l** Ectopic YTHDF2 expression rescues SVIL or MLL1 knockdown-induced
growth inhibition in LFS astrocytes (n = 6 biologically independent samples). **m** In
vitro AIG assay demonstrates decreased colony numbers upon SVIL or MLL1
knockdown that are rescued by YTHDF2 expression (n = 3 biologically independent
samples)). **n** Colony-forming assay demonstrates that OICR-9429 and MI-2-2 cause
more severe growth inhibition of LNZ308-p53(G245D) cells than LNZ308-Vector
cells (n = 3 biologically independent samples). **o** Images of WT (mCherry+) and LFS
(GFP+) co-cultured cerebral organoids examined by light sheet fluorescence
microscopy. Scale bar, 500 μm. **p** OICR-9429 selectively inhibits proliferation of the
LFS-derived population of WT/LFS co-cultured cerebral organoids (n = 19 biologi-
cally independent samples). The results are representative of at least three inde-
pendent experiments (**c–f, i, o**). The data are presented as the mean ± SEM; two-way
ANOVA with Bonferroni's multiple comparison test (**h, j–n**); unpaired two-tailed
Student's *t* test (**p**); ***P < 0.001. ns not significant. Source data and exact *P* values
are provided in the Source Data file.

## YTHDF2-mediated *CDKN2B* and *SPOCK2* mRNA degradation contributes to mutant p53-associated malignancy

To examine whether the mutant p53-rewired YTHDF2-associated m⁶A epitranscriptomic networks contribute to mutant p53 gain-of-function toward glioma progression in vivo, we investigated the role of the mutant p53/SVIL/MLL1/YTHDF2 axis in LNZ308-p53(G245D) tumor-igenesis. Xenograft studies revealed that depletion of either SVIL, MLL1, or YTHDF2 profoundly antagonizes tumorigenesis of LNZ308-p53(G245D) glioma cells, suggesting a critical oncogenic role of the mutant p53/SVIL/MLL1 complex by activating YTHDF2 to promote tumorigenesis in vivo (Fig. 6a).

The m⁶A-marked *CDKN2B* and *SPOCK2* transcripts were negatively regulated by mutant p53/SVIL/MLL1-regulated YTHDF2 in LFS astro-cytes (Fig. 5g, h, k, l), suggesting that suppression of CDNK2B and SPOCK2 may promote tumor initiation. We next assessed the role of YTHDF2 targets CDKN2B and SPOCK2 on mutant p53-induced glio-magenesis. Ectopic expression of CDKN2B or SPOCK2 significantly suppressed the cell proliferation and in vitro AIG ability of LFS astro-cytes (Fig. 6b, c). Xenograft assay revealed that ectopic expression of CDKN2B or SPOCK2 profoundly inhibits LNZ308-p53(G245D) tumor-igenesis (Fig. 6d). These results establish that YTHDF2-mediated *CDKN2B* and *SPOCK2* mRNA degradation contributes to mutant p53 oncogenic activities. Moreover, the knockdown of CDKN2B or SPOCK2 remarkably rescued impaired cell proliferation and colony growth of YTHDF2-depleted LFS astrocytes (Fig. 6e, f). In vivo tumorigenesis assays revealed that the depletion of CDKN2B or SPOCK2 profoundly antagonized the loss of YTHDF2-induced suppression of LZN308-p53(G245D) (Fig. 6g). Together, our results confirm that YTHDF2-suppressed *CDKN2B* and *SPOCK2* transcripts play a role in mutant p53-mediated oncogenic features in LFS cells and p53-mutated gliomas.

Loss of the CDKN2B tumor suppressor is one of the most common genomic alterations in malignant glioma[41], emphasizing the critical role of CDKN2B in gliomagenesis. We examined transcriptome-wide effects of altered CDKN2B expression by restoring CDKN2B in LFS astrocytes. KEGG and GO_BP analyses indicated that CDKN2B restoration downregulates cell cycle and DNA replication pathways (Fig. 6h). Of note, these pathways were enriched in LFS astrocytes (Supplementary Fig. 7a–c). These findings emphasize the critical role of YTHDF2-mediated CDKN2B downregulation in mutant p53-initiated neoplastic transformation in LFS astrocytes.

## Clinical relevance of mutant p53-regulating YTHDF2 in LFS and human cancers

To explore the clinical relevance of our findings, we first examined YTHDF2 and YTHDF2-regulated *CDKN2B* mRNA expression in stromal tissue of healthy people and LFS patients[42]. Compared with healthy donor stroma, higher YTHDF2 expression and lower CDKN2B expression were observed in LFS patients bearing an inherited p53 missense

mutation but not in LFS patients with an inherited p53 frameshift mutation (Fig. 7a). These findings from clinical samples indicate that specific mutant p53s develop a gain-of-function that activates the YTHDF2/CDKN2B regulatory axis.

To determine whether our findings from LFS-derived astrocytes could be translated to human primary tumors with somatic p53 mutations, we next comprehensively analyzed the association between p53 mutation status and YTHDF2 expression using TCGA databases for 4 cancer types (low grade glioma (LGG), GBM, breast invasive carcinoma (BRCA), and rectal adenocarcinoma (READ)) characterized by high p53 mutation rates. Tumor specimens were grouped into p53 WT (no detectable p53 mutation) and p53 missense mutations, with tumor samples bearing p53 frameshift or nonsense mutations excluded. Compared to the p53 WT group, tumors with p53 missense and hotspot mutations demonstrated lower expression of p53 targets p21 and PUMA, but retained high YTHDF2 expression in LGG, GBM, BRCA and READ cancers (Fig. 7b and Supplementary Fig. 8a). Of note, p53 mutation status was also positively correlated with high expression of SOX2, an undifferentiated glioma stem cell marker (Supplementary Fig. 8b). These clinical cancer studies emphasize the general regulatory mechanism of mutant p53 in YTHDF2 expression.

We next examined the clinical correlation of YTHF2 with *CDKN2B* and *SPOCK2* transcripts. The expression of YTHDF2 was negatively correlated with *CDKN2B* and *SPOCK2* mRNAs in multiple pediatric glioma datasets (Supplementary Fig. 8c, d), supporting the patholo-gical role of YTHDF2 negatively regulating *CDKN2B* and *SPOCK2* tran-scripts in glioma development. To study the clinical significance of the YTHDF2-regulated epitranscriptome in human cancers, we examined the association of YTHDF2-regulated targets with patient prognosis. A YTHDF2 negatively regulated gene signature was defined as the 20 genes whose m⁶A-markings increased in LFS astrocytes compared to WT astrocytes but decreased in LFS astrocytes upon YTHDF2 deple-tion. As expected, lower expression of YTHDF2 targets was correlated with significantly decreased LGG/BGM patient survival (Fig. 7c). Importantly, expression of numerous YTHDF2 targets (13 out 20, including CDKN2B and SPOCK2) was also negatively associated with survival hazard ratio in LGG/BGM patients (Fig. 7d). Taken together, our findings demonstrate that mutant p53-regulated YTHDF2 and YTHDF2-mediated m⁶A mRNA degradation play a critical role in glioma development and could serve as potentially targetable vulnerability.

## Discussion

Patient-derived iPSCs present a powerful cancer platform to assess the cellular signaling, transcriptional, and chromatin landscapes resulting from well-defined genetic alterations and provide potentially ther-apeutic insights into the early events of tumor initiation[26,43–48]. Onco-genic activities of mutant p53s have been demonstrated in human

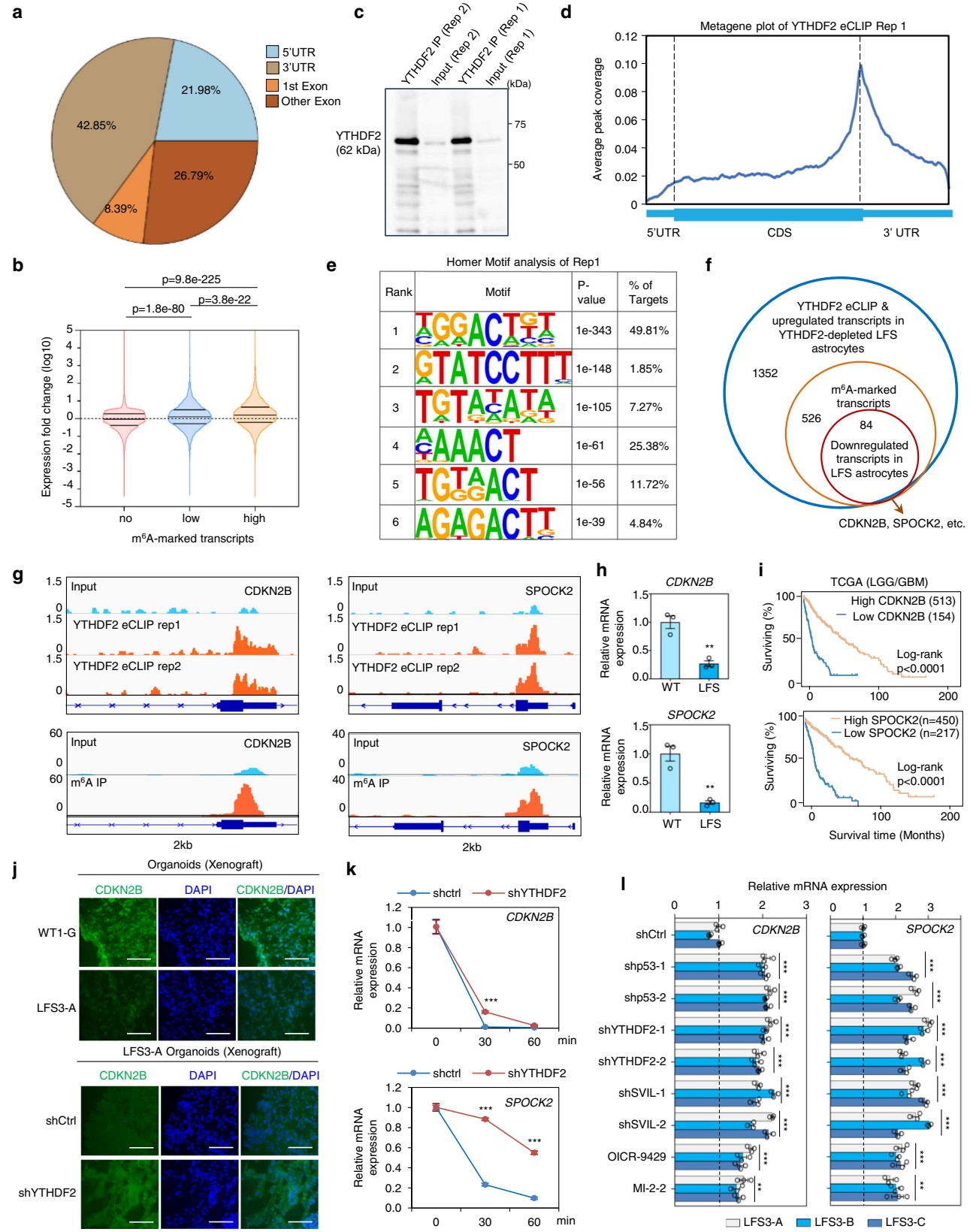

cancers throughout tumor initiation, progression, metastasis, and recurrence[49,50]. Our study, utilizing a LFS iPSC disease platform to investigate oncogenic events in LFS patients, helps to dissect the pathological mechanisms triggered by mutant p53 at the earliest stage of glioma development. We demonstrate that p53 missense mutations (e.g., R175H and G245D) help form a unique mutant p53/SVIL/MLL1

complex. MLL1 forms a COMPASS-like complex together with WDR5, MENIN, RBBP5, and ASH2L, which introduce H3K4me3 marks on gene promoters to facilitate mRNA Pol II-mediated YTHDF2 transcription[51,52]. Small-molecule inhibitors OICR-9429 and MI-2-2 were designed to inhibit MLL1 function by disrupting the interaction between MLL1 and COMPASS-like complex members WDR5 and

**Fig. 5 | Identification of YTHDF2 targets via m⁶A MeRIP-seq, eCLIP-seq, and RNA-seq in LFS astrocytes. a** m⁶A MeRIP-seq indicates distribution of m⁶A peaks in different regions (5'UTR, first exon, other exon, and 3'UTR) of transcripts. Pie chart shows the percentage of m⁶A peaks within distinct regions of transcripts in LFS astrocytes. **b** Violin plot demonstrates significant elevation of highly m⁶A-marked transcripts upon YTHDF2 depletion. **c** Examination of YTHDF2 IP enrichment by eCLIP-seq. **d** Metagene plot of YTHDF2 eCLIP-seq indicates enrichment of YTHDF2-interacting mRNA peaks in the 3'UTR clustered around stop codons. **e** Motif analysis demonstrates that YTHDF2 binding motifs are similar to the consensus m⁶A motif RRACH. **f** Venn diagram identifying 84 YTHDF2-targeted m⁶A transcripts validated by a combination of m⁶A MeRIP-seq, YTHDF2 eCLIP-seq, and RNA-seq in LFS astrocytes. These transcripts include *CDKN2B* and *SPOCK2* mRNAs. **g** IGV track views of m⁶A peaks located on *CDKN2B* and *SPOCK2* transcripts in LFS astrocytes. **h** RT-qPCR indicates decreased expression of YTHDF2-targeted *CDKN2B* and *SPOCK2* transcripts in LFS astrocytes (*n* = 3 biologically independent samples). **i** Low expression of YTHDF2-targeted CDKN2B and SPOCK2 is correlated with poor overall survival of LGG/GBM patients. Log-rank (Mantel–Cox) test is performed to compute significance. **j** Immunostaining demonstrates lower CDKN2B in engrafted LFS cerebral organoids (upper panel) and increased CDKN2B in YTHDF2-depleted engrafted LFS cerebral organoids (lower panel), Scale bar, 100 μm. Anti-CDKN2B antibodies only recognize human but not mouse CDKN2B proteins. **k** mRNA stability assay demonstrates that YTHDF2 knockdown leads to an increased half-life of *CDKN2B* and *SPOCK2* mRNAs. shCtrl-LFS and shYTHDF2-LFS astrocytes are treated with actinomycin D and total RNAs are isolated at 0, 30, and 60 min. (*n* = 3 biologically independent samples). **l** RT-qPCR demonstrates elevated expression of YTHDF2 targets CDKN2B and SPOCK2 upon depletion of p53, YTHDF2, or SVIL as well as inhibition of MLL1 function by OICR-9429 or MI-2-2 in LFS astrocytes (*n* = 3 biologically independent samples). The results are representative of at least three independent experiments (**c, j**). The data are presented as the mean ± SEM); two-way ANOVA with Bonferroni's multiple comparison test (**l**); unpaired two-tailed Student's *t* test (**h**); multiple *t* test (**k**); \*\**P* < 0.01, \*\*\**P* < 0.001. Source data and exact *P* values are provided in the Source Data file.

MENIN, respectively[53,54], highlighting the essential role of the COMPASS-like complex in MLL1-mediated H3K4me3. This work establishes that increased YTHDF2 hijacks the epitranscriptomic regulatory network by promoting m⁶A-mediated degradation and silencing of an array of tumor suppressor genes, culminating in neoplastic transformation (Fig. 7e). Importantly, this regulation not only contributes to early gliomagenesis in LFS patients but also in glioma cell lines and clinical glioma specimens with p53 missense mutations.

CDKN2B loss is an important mechanism for tumors to bypass cell cycle checkpoints. Clinically, homozygous deletion of the *CDKN2B* gene is observed in 30% of gliomas[41], supporting the link between impaired CDKN2B activity and gliomagenesis. However, it remains unknown whether mutant p53 may also modulate CDKN2B function to promote gliomagenesis. Interestingly, CDKN2B is downregulated by YTHDF2-mediated m⁶A *CDKN2B* mRNA degradation in premalignant LFS astrocytes with intact genomic *CDKN2B*, suggesting that mutant p53 alone can bypass oncogene-induced senescence by inhibiting CDKN2B[55]. This characteristic CDKN2B downregulation may be useful in the future as both a biomarker to identify and a therapeutic strategy to treat early-stage gliomas in LFS patients.

Multiple groups have demonstrated that mutant p53 gain-of-function mutations may alter the protein-protein interaction (PPI) network and recruit a unique complement of transcription factors (e.g., ETS2[56], NRF2[57], and SREBP[6]) and chromatin complexes[58] to modulate gene expression. Mutant p53s can also bind to promoters and lead to transactivation of downstream genes[8]. It is conceivable that phenotypic differences observed from different p53 missense mutants result from their distinct functional binding protein partners and genome occupancies.

In contrast to the direct interactions of mutant p53 with transcriptional cofactors p300 or PCAF acetyltransferases reported to drive a unique genomic binding pattern and target gene expression[59,60], our ChIP-seq studies demonstrated that both p53 and mutant p53 have a similar genome-binding affinity on the *YTHDF2* promoter. However, mutant p53 but not p53 recruits SVIL to form a functional complex with MLL1 to transactivate YTHDF2 expression and promote cell proliferation and neoplastic transformation, highlighting this important difference between genomic occupancy and transcriptional activation. These findings establish that distinct mutant p53s drive diverse biological behavior through multiple layers of gene regulation. Interestingly, a previous report demonstrated that MLL1 is a transcriptional target of mutant p53s (e.g., p53(R248Q), p53(R249S), and p53(R273H)) but not WT p53 in breast cancers[8]. Our studies suggest that MLL1 is not only a direct target of mutant p53s, but also acts as an epigenetic co-factor to drive mutant p53-mediated gene activation by depositing H3K4me3 on the YTHDF2 promoter. Our work also establishes that YTHDF2 activation leads to initiation of gliomagenesis in LFS patients. Taken together, our investigations of mutant p53-induced glioma signatures reveal a subtle layer of transcriptional and epigenetic regulation driving oncogenesis.

Dysregulation of m⁶A modification and m⁶A-associated regulators likely plays a critical role in the initiation and progression of brain tumors[21,61], and is supported by the evidence of reduced m⁶A levels in gliomas and inhibition of self-renewal of glioma stem cells (GSCs) by m⁶A writer METTL3 methylation of the *ADAM19* transcript[24,62]. In addition, the m⁶A eraser ALBKH5 promotes GBM invasiveness by demethylating the *FOXM1* transcript[63]. Our studies conclude that YTHDF2 is capable of regulating neoplastic transformation and cell proliferation among astrocytes by degrading tumor suppressor transcripts including CDKN2B. These findings indicate that decreased m⁶A RNA methylation, which can be determined by the balance between the diverse functions of m⁶A writers, erasers, and readers, leads to gliomagenesis. Importantly, mutant p53 is capable of regulating global m⁶A profiles by transcriptionally activating expression of the m⁶A reader YTHDF2. Overall, this suggests that mutant p53 plays a critical role in altering the m⁶A-associated epitranscriptomic landscape to initiate gliomagenesis in LFS patients as well as promote tumorigenesis in p53-mutated gliomas. Future studies dissecting the m⁶A epitranscriptomic regulatory network in gliomagenesis may identify additional therapeutic opportunities for interrupting this process.

Transcription factors including mutant p53s are not considered to be suitable druggable targets for cancer therapies; therefore, mutant p53-regulated oncogenic targets and their associated regulatory machinery will likely be the most exploitable therapeutic vulnerabilities for cancer treatment and prevention among LFS patients. In addition to accumulating evidence demonstrating the beneficial therapeutic effect of blocking the m⁶A epitranscriptomic landscape in GSCs[63], our study further emphasizes the potential for therapeutic strategies preventatively targeting mutant p53/SVIL/MLL1-upregulated YTHDF2 expression in LFS patients as well as treating gliomas harboring p53 missense mutations.

## Methods

This research complies with all relevant ethical regulations approved by The University of Texas Health Science Center at Houston Institutional Biosafety Committee and Human Embryonic Stem Cell Research Oversight Committee. The animal studies are approved by the Institutional Animal Welfare Committee.

### WT and LFS iPSC generation and cell culture

Patient-derived WT and LFS iPSCs were generated using the non-integrative Sendai virus-4F reprogramming method, as reported previously[26]. Four independent WT iPSC lines (WT1-G, WT1-H, WT1-J, and WT1-K) and five independent LFS iPSC lines (LFS3-A, LFS3-B, LFS3-C, LFS3-D, and LFS3-E) generated from fibroblasts from the mother and father in the LFS family pedigree were utilized in this study. H1

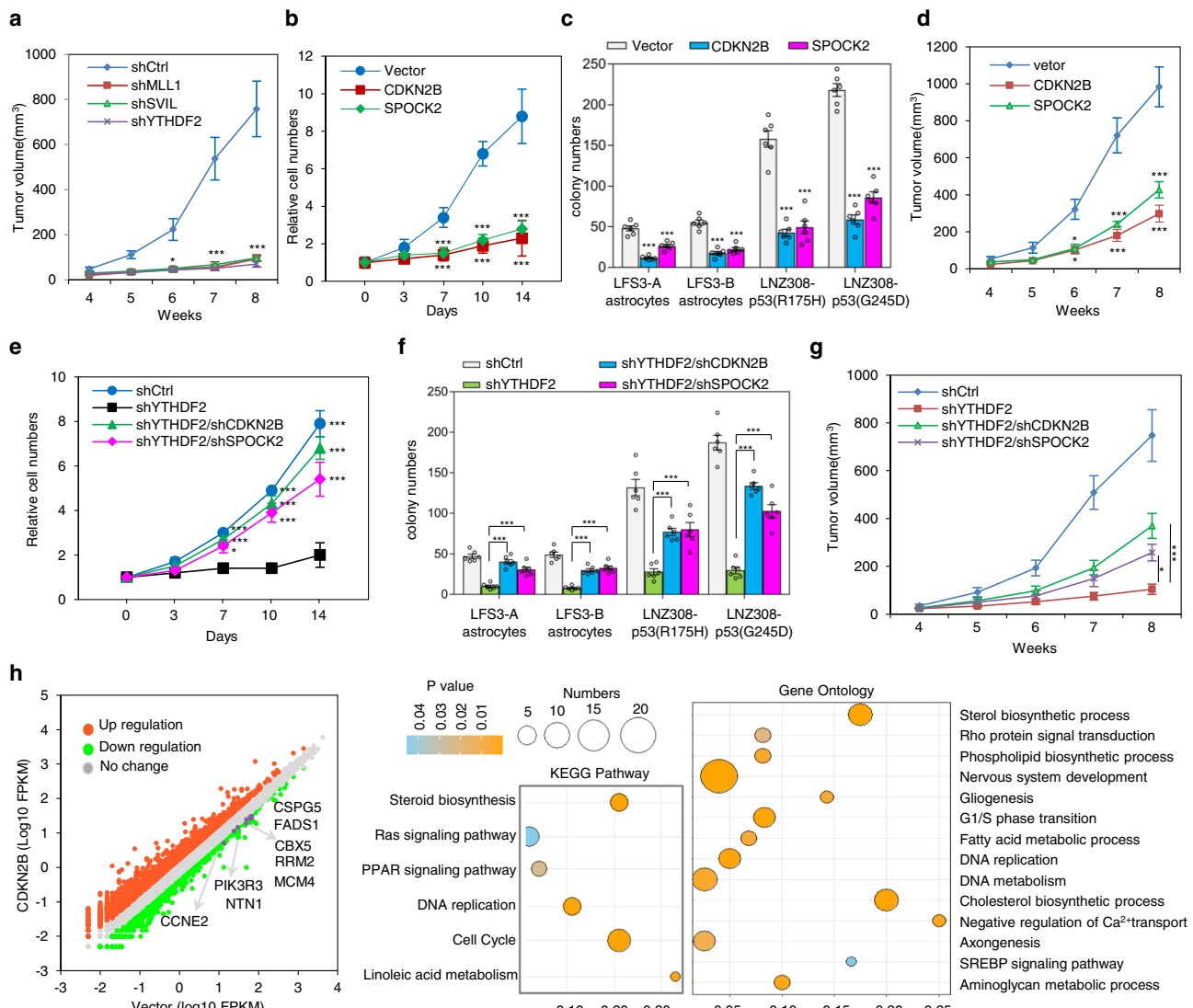

**Fig. 6 | YTHDF2-mediated *CDKN2B* and *SPOCK2* mRNA degradation contributes to mutant p53-associated malignancy. a** In vivo mouse xenograft study demonstrates that knockdown of YTHDF2, SVIL, or MLL1 hampers LNZ308-p53(G245D) tumor growth (*n* = 4 biologically independent mice). The sizes of the tumors were measured at the indicated time. **b** Expression of CDKN2B or SPOCK2 inhibits cell proliferation of LFS astrocytes (*n* = 6 biologically independent samples). **c** In vitro AIG assay demonstrates ectopic expression of CDKN2B or SPOCK2 leading to decreased colony numbers of LFS astrocytes. All colonies are counted and measured after 2-month culture (*n* = 6 biologically independent samples). **d** In vivo mouse xenograft study shows ectopic expression of CDKN2B or SPOCK2 abrogating LNZ308-p53(G245D) tumor growth (*n* = 3 biologically independent mice). The sizes of the tumors were measured at the indicated time. **e** Knockdown of CDKN2B or SPOCK2 rescues YTHDF2 depletion-induced growth inhibition in LFS astrocytes (*n* = 6 biologically independent samples). **f** In vitro AIG assay demonstrates knockdown of CDKN2B or SPOCK2 rescuing in vitro colony formation of YTHDF2-depleted LFS astrocytes. All colonies are counted and measured after 2-month culture (*n* = 6 biologically independent samples). **g** Xenograft study indicates that knockdown of CDKN2B or SPOCK2 rescues YTHDF2 knockdown-induced LNZ308-p53(G245D) tumor growth inhibition in nude mice (*n* = 4 biologically independent mice). The sizes of the tumors were measured at the indicated time. **h** Transcriptome analysis of CDKN2B-restored LFS astrocytes. Bubble plot for visualizing enriched GO and KEGG pathway analyses of differentially upregulated genes in CDKN2B-restored LFS astrocytes. *X* axis label represents the enrichment factor (number of differentially expressed genes enriched in the pathway/total number of genes in the pathway) and *Y* axis label represents GO annotation and KEGG pathway. Size and color of the bubble represent number of differentially expressed genes enriched in the GO or KEGG pathways and enrichment significance (*P* value), respectively. The data are presented as the mean ± SEM; two-way ANOVA with Bonferroni's multiple comparison test (**a**–**g**). **P* < 0.05, ****P* < 0.001. Source data and exact *P* values are provided in the Source Data file.

hESCs (WA01) were purchased from WiCell. HEK-293 and HEK-293T cells were obtained from the American Type Culture Collection (ATCC). and glioma cell line LZN308 was provided by Dr. Jova Chandra at MD Anderson Cancer Center. Both iPSCs and H1 hESCs were maintained in mTeSR1 medium (Stemcell Technologies) or DMEM/F12 supplemented with 20% KnockOut serum replacement and 10 ng/mL bFGF on Matrigel-coated plates. HEK-293, HEK-293T, and LNZ308 cells were cultured and maintained in DMEM supplemented with 10% (vol/vol) fetal bovine serum (FBS) (F0900-050, GenDEPOT), L-glutamine, non-essential amino acids, β-mercaptoethanol, and penicillin/streptomycin. LFS iPSCs and hESCs were approved by the Human Embryonic Stem Cell Research Oversight Committee, The University of Texas Health Science Center at Houston, Houston, TX, USA (SCRO-20-03 and SCRO-20-07).

**Animal models.** All animal procedures used in this study were performed under protocols approved by the institutional Animal Welfare Committee, Stem Cell Research Oversight Committee, Biosafety Committee, and Committee for the Protection of Human Subjects of the University of Texas Health Science Center at Houston. Eight-week-old

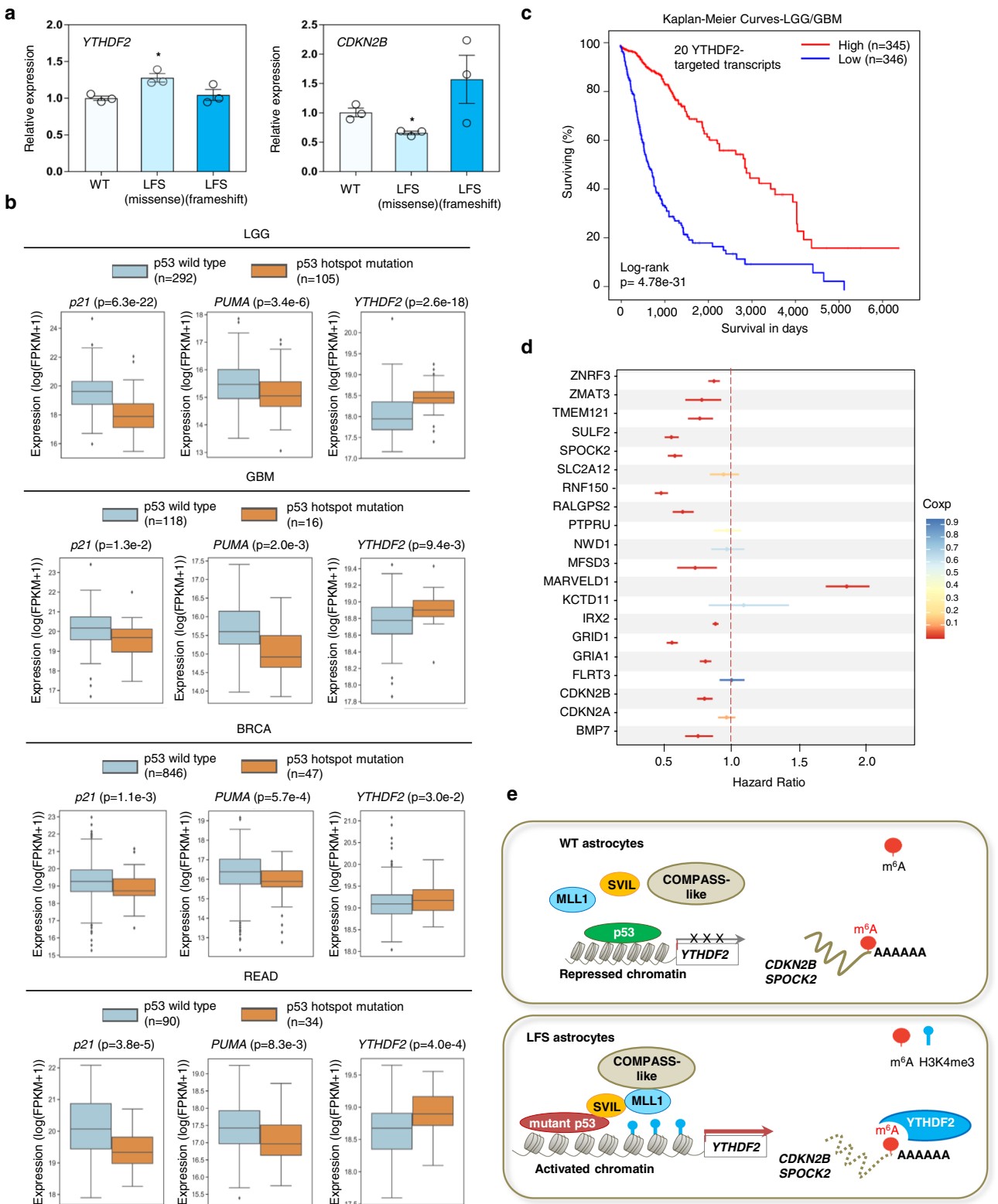

outbred athymic nude (*Foxn1^nu*) mice (The Jackson Laboratory, RRID: IMSR JAX:007850) were used for all experiments and were randomly assigned to treatment groups. Animals were monitored every day for 3 days after surgery and at least twice weekly for physical and neurological abnormalities.

**Plasmids, shRNAs, antibodies, and chemicals.** The constructs for CDKN2B ectopic expression were generated by inserting CDKN2B

cDNAs, respectively, into the TetO-FUW-8MCS vector using the EcoRI and BstB1 restriction enzyme cutting sites. The lentiviral shRNAs were designed in the TRC library database (https://portals.broadinstitute. org/gpp/public/) and inserted into either pLKO.1 (Addgene, 8453) or pLKO.pig, which was described previously[64,65].

The Flag-YTHDF2 was a gift from Dr. Markus Bohnsack (University Medical Center, Germany)[66]. The GFP-SVIL was a gift from Drs. Elizabeth J. Luna (University of Massachusetts Medical School, USA) and

**Fig. 7 | Clinical relevance of mutant p53 in regulating YTHDF2 expression and the prognostic value of YTHDF2 targets in glioma patients. a** Elevated YTHDF2 expression is observed in LFS stromal cells with a heterozygous M133T mutation but not LFS stromal cells with a heterozygous 12141delG frameshift mutation compared with WT stroma (n = 3 biologically independent samples). The data are presented as the mean ± SEM; unpaired two-tailed Student's t test. *P < 0.05. **b** Box plots of TCGA RNA expression profiles (log2) in TCGA tumors with p53 WT or p53 hotspot missense mutations in LGG, GBM, BRCA, and READ specimens. Two-sided Mann Whitney Wilcoxon test is performed to compute significance. Tumors with a p53 hotspot missense mutation demonstrate decreased *p21* and *PUMA* mRNA expression but elevated *YTHDF2* mRNA expression. Box edges delineate lower and upper quartiles, the center line represents the median, and whiskers extend to 1.5 times the interquartile range. **c** Kaplan–Meier curves compare survival in LGG and GBM specimens with high or low levels of YTHDF2-targeted transcripts. Log-rank

(Mantel–Cox) test is performed to compute significance. **d** Estimated hazard ratios and 95% confidence intervals for TCGA LGG and GBM patients expressing high levels of 20 YTHDF2-targeted genes. High expression of 13 of these 20 genes is positively associated with lower hazard ratios and increased survival with FDR q-value less than 5%. **e** A model linking the mutant p53/SVIL/MLL1 transcriptional regulatory complex to epitranscriptomic changes driving gliomagenesis. In our proposed model, mutant p53 interacts with SVIL, recruits MLL1 to the YTHDF2 promoter, and then induces YTHDF2 transcription. Elevated YTHDF2 down-regulates numerous m6A-marked transcripts, including *CDKN2B* and *SPOCK2*, promotes neoplastic transformation and initiates gliomagenesis. MLL1 inhibitors selectively suppress YTHDF2 expression, LFS and mutant p53 cell survival, and neoplastic transformation. Source data and exact P values are provided in the Source Data file.

Zhiyou Fang (The Hefei Institutes of Physical Science of the Chinese Academy of Sciences, China). The pLenti6/V5-p53(R175H) and pLenti6/V5-p53(G245D) were Lee lab stocks and described previously[26]. shRNAs and RT-qPCR primers to validate knockdown efficiency used in this study are listed in Supplementary Data 6.

Antibodies against p53 (1:1000, Santa Cruz, sc-126), SOX2 (1:200, Santa Cruz, sc-17320), SOX10 (1:200, Santa Cruz, sc-17342), OLIG2 (1:200, R&D Systems, BAF2418), GFAP (1:200, BioLegend, 837201), β-TUBULIN III (1:200, Sigma-Aldrich, SAB4200715), NESTIN (1:200, Bio-Legend, 655102), Ki67 (1:200, Life Technologies, 14-5698-82), VCL (1:2000, Sigma-Aldrich, V4505), PAX6 (1:200, Biolegend, 901301), Flag (1:2000, Sigma-Aldrich, F1804), HA (1:2000, Roche, 11666606001), V5 (1:2000, Thermo Fisher Scientific, R960-25), GFP (1:2000, Santa Cruz, sc-9996), m6A (1:1000, Synaptic Systems, 202003), YTHDF2 (1:1000, Proteintech, 24744-1-AP; 1:200, Aviva, ARP67917_P050), SVIL (1:500, Sigma-Aldrich, S8695), STEM121 (1:200, Takara Bio, Y40410), human nuclear antigen (1:200, Novus Biologicals, NBP2-34342), MLL1 (1:500, Bethyl Laboratories, A300-086A), H3K4me3 (1:200, Abcam, ab8580), H3K27Ac (1:200, Abcam, ab4729), METTL3 (1:1000, Proteintech, 15073-1-AP), METTL14 (1:1000, Cell Signaling, 48699), WTAP (1:1000, ABclonal Technology, A14695), FTO (1:1000, ABclonal Technology, A3851), ALKBH5 (1:1000, ABclonal Technology, A11684), CDKN2B (1:200, Thermo Fisher Scientific, MA1-12294), and SPOCK2 (1:200, Bioss Antibodies, BS-11966R) were purchased from the indicated suppliers. Chemicals OICR-9429 (Tocris, 5267), MI-2-2 (Sigma-Aldrich, 444825), and Y-27632 (Stemcell Technologies, 72304) were purchased from the indicated suppliers.

**In vitro differentiation of iPSCs and hESCs to NPCs, and then to astrocytes.** iPSCs and hESCs were cultured on Matrigel growth factor reduced basement membrane matrix (Corning, 35423)-coated plates in mTeSR1 medium (Stemcell Technology, 85850) and maintained at 37 °C in a humidified 5% $CO_2$ incubator. iPSCs were passaged after reaching 85% confluence. The plates were pre-coated with 1:50 diluted Matrigel at room temperature for 1 h. mTeSR1 medium with ROCK inhibitor Y-27632 was used to improve the survival of dissociated hESCs. The culture medium was changed every other day. Cells were passaged every 5–7 days at a 1:10 ratio.

NPC and astrocyte differentiation were performed using modified neural induction methods as described by the Neural Induction kit manufacturer (Stemcell Technologies) (Supplementary Fig. 1b). Briefly, embryoid bodies (EBs) were formed from iPSCs and hESCs by treatment with the STEMdiff SMADi Neural Induction kit (0.1 μM LDN193189 and 10 μM SB431542) in AggreWell 800 plates (Stemcell Technologies, 34811). On day 5, EBs were collected and replated onto a Matrigel-coated well of a six-well plate, and then maintained in NPC culture medium. Rosette selection was performed after 14 days of culture by STEMdiff Neural Rosette Selection Reagent (StemCell Technologies, 05832). iPSC and hESC-derived NPCs were maintained in STEMdiff Neural Progenitor Medium (Stemcell Technologies,

05833) and validated by immunostaining with NPC markers (SOX2, PAX6, and NESTIN) and frozen in liquid N2 for future usage. For astrocyte differentiation, NPCs were maintained at high density on Matrigel in STEMdiff Neural Progenitor Medium. NPCs were then dissociated into single cells and seeded at 15,000 cells/cm² density on Matrigel-coated plates in the astrocyte medium (ScienCell Research Laboratories, 1801). After 20 days of culture, astrocytes were split 1:3 every week with Accutase cell detachment solution (StemCell Technologies, 07922), continually cultured until day 75, and validated by immunostaining with NPC markers (S100β and GFAP).

**Co-IP and immunoblotting.** Co-IP and immunoblotting assays were performed as previously described[64,67,68]. The uncropped and unprocessed scans of all blots are provided in the Source Data file.

**m6A dot blotting.** mRNA was extracted from astrocytes and cell lines by mRNA isolation kit (Sigma-Aldrich, 11741985001) according to the manufacturer's instructions and quantified by a NanoDrop 2000 spectrophotometer (Thermo Fisher Scientific). mRNA samples were diluted to a concentration of 2 ng/μl using RNase-free water. Diluted mRNA was denatured at 95 °C for 3 min to disrupt secondary structures. Samples were quickly placed on ice for 5 min and 2 μl of mRNA from each sample was dropped directly onto the Amersham Hybond-N + membrane (GE Healthcare Life Science). mRNA blots were crosslinked to the membrane using a UV crosslinker at $1.2 \times 10^5$ μJ for 30 s. The membrane then was washed at room temperature with PBST buffer (PBS buffer with 0.1% Tween 20) for 5 min. The membrane was incubated in 10 ml of blocking buffer (PBST buffer with 5% skim milk) for 1 h, followed by overnight incubation with anti-m6A antibody (1:250 dilution) with gentle shaking. The membrane was incubated with the secondary antibody for 1 h and detected by immunoblotting with the Pierce ECL Western blotting substrate. Non-denatured mRNAs were loaded as a loading control and stained with 0.02% methylene blue in 0.3 M sodium acetate (pH 5.2) overnight at 4 °C. The relative signal density of each dot was measured by ImageJ.

**In vitro AIG and colony-formation assays.** The AIG assay was performed as described previously[26]. Briefly, WT and LFS iPSC-derived astrocytes were split and suspended in the astrocyte medium with 0.3% SeaPlaque low-melting agarose (Lonzo, 50100). $1 \times 10^4$ cell suspensions were plated into each well of six-well plates containing solidified 0.8% agarose in the astrocyte medium. Cells were maintained for 1 month with 1.5 ml astrocyte medium per well. Half of the medium was replaced every 3 days. Colonies were stained by crystal violet staining solution (0.5% crystal violet (Sigma-Aldrich, C6158) in 25% methanol) for clear colony visualization. All colonies visible under a Leica DMi8 microscope with 5X objective were counted in nine random-pick views for each group. Colony-formation assay was performed as described previously[69]. Briefly, LNZ308-Vector and LNZ308-p53(G245D) cells were split and seeded at $1 \times 10^3$ cells per well in six-well plates with

complete DMEM medium (DMEM supplemented with 10% FBS, 2mM L-Glutamate, and 100units/ml Penicillin-Streptomycin). The medium was changed to complete DMEM supplemented with DMSO, OICR-9429, or MI-2-2 on day 1, which was replaced on day 4. After 7 days of culture, colonies were stained with 0.005% crystal violet for 15 min and then washed with tap water at room temperature. Colony numbers were analyzed by ImageJ.

**In vitro competition assay.** In vitro competition assay was performed as described previously[26,65]. Briefly, WT astrocytes were labeled with mCherry by infecting them with a mCherry lentivirus, while LFS astrocytes were labeled by GFP by infecting them with a GFP lentivirus. GFP+ and mCherry+ cells were mixed at a 50:50 ratio and confirmed by a BD LSR II flow cytometer (BD Biosciences) at day 0 (P0). Mixed cells were maintained until confluence. After confluence, cells were split, passaged, and examined for the GFP+/mCherry+ population by a flow cytometer for 3 passages (P1, P2, and P3). Gates drawn were based on cell populations between negative (isotype controls) and positive (single-staining) staining. The flow cytometry data were analyzed by FlowJo v8.

**RT-qPCR.** Total RNA was isolated from iPSC-derived astrocytes and cell lines using TRIzol reagent (Invitrogen, 15596026) according to the manufacturer's instructions. Overall, 1 μg RNA was used for reverse transcription using the iScript cDNA synthesis kit (Bio-Rad Laboratories, 1708891). RT-qPCR reaction was performed using a CFX96 machine (Bio-Rad Laboratories). A 20 μl RT-qPCR reaction solution was composed of 10 μl SYBR Green PCR Master Mix (Bio-Rad Laboratories, 1725124), 1 μl cDNA, 1 μl each of 10 μM forward and reverse qPCR primers, and 7 μl RT-PCR grade water. All reactions were run in at least triplicate and normalized to GAPDH expression. Primer sequences are listed in Supplementary Data 6.

**RNA-seq.** Cell samples were lysed in the TRIzol reagent. RNA-seq was performed by either UTHealth Houston Cancer Genomics Center or BGI Genomics. All RNA-seq data analyses were performed using Galaxy Community Hub (https://galaxyproject.org/) to calculate FPKM (Fragments Per Kilobase of transcript per Million) as described previously[26,46,47].

**ChIP-seq and ChIP-PCR analysis.** ChIP was performed using modified previous methods[46]. For p53 and H3K27Ac, cells were fixed in 1% formaldehyde (Thermo Fisher Scientific, 28906) at room temperature for 10 min. After glycine quenching, cell pellets were collected and lysed and then subjected to sonication using a Branson Sonifier 450 (sonication conditions: 10 s on and 10 s off for 45 min) on ice. The supernatant was then diluted in the same sonication buffer, and subjected to immunoprecipitation with corresponding antibodies against p53 or H3K27Ac at 4 °C overnight. The Protein A/G Dynabeads were added and incubated at 4 °C for 2–3 h, washed with high alt buffer and TE buffer, and then eluted with TE buffer with 1% SDS at 4 °C for 15 min. Following ChIP, DNA was quantified by qPCR using standard procedures and ChIP-seq libraries were prepared using KAPA HyperPrep Kit (Roche, KK8502). ChIP-seq was performed on an Illumina NextSeq 550 platform at the UTHealth Houston Cancer Genomics Center. ChIP-qPCR was performed using a CFX96 machine (Bio-Rad Laboratories). The 20 μl RT-qPCR solution was composed of 500 ng ChIP product, 1 μl respectively of 10 μM forward and reverse PCR primers for amplifying the peak region, 10 μl SYBR Green PCR Master Mix (Bio-Rad Laboratories, 1725124), and 7 μl DNase/RNase-free water. Primer sequences are listed in Supplementary Data 6.

ChIP-seq analyses were performed using Galaxy Community Hub (https://galaxyproject.org/). Briefly, the sequencing reads were processed by Trim Galore! (https://www.bioinformatics.babraham.ac.uk/projects/trim_galore/) for quality control and then aligned to the human reference genome (hg19) using Bowtie2 (ref. [70]). PCR duplicates were removed by SAMtools RmDup[71] before peak-calling was performed using MACS2 (ref. [72]). The coverage bigWig files were generated using deepTools/bamCoverage[73] and visualized on IGV browser[74]. Heatmaps were generated using deepTools/computeMatrix and plotHeatmap[73]. The genomic annotations and motif analyses were performed using HOMER (http://homer.ucsd.edu/homer/index.html)[75].

**Electrophoretic mobility shift assay.** Fluorescence-based electrophoretic mobility shift assay (EMSA) was performed according to the manufacturer's manual (Thermo Fisher Scientific, E33075). Briefly, YTHDF2 promoter DNA was incubated with indicated proteins in the binding buffer at room temperature for 20 min and then separated on a 4–20% TBE gel (Thermo Fisher Scientific, EC6225BOX). The gel was stained with SYBR Green EMSA stain and the images were captured using a Bio-Rad ChemiDoc MP imaging system.

**RNA stability assay.** LFS astrocytes (shCtrl and shYTHDF2) were treated with actinomycin D (5 μg/mL, Sigma-Aldrich, A9415) for the indicated duration and then harvested using the TRIzol reagent (Invitrogen, 15596026) for total RNA extraction. Equal amounts of total RNA for each sample were subjected to RT-qPCR to measure CDKN2B and SPOCK2 mRNA levels.

**Enhanced cross-linking and immunoprecipitation (eCLIP).** Experiments were performed as previously described[39]. Briefly, LFS astrocytes were UV crosslinked and then sonicated in lysis buffer on ice. An antibody against YTHDF2 (Aviva, ARP67917_P050) was used for immunoprecipitation and 2% input was collected to run alongside IP samples. RNA-protein complexes were run on SDA-PAGE and transferred to nitrocellulose membranes. RNA between 65 kDa to 140 kDa was excised from the membrane, treated with proteinase K (NEB), and reverse transcribed for library construction. Sequencing was performed on an Illumina NextSeq 550 platform at the UTHealth Houston Cancer Genomics Center. Data analysis was run through the eCLIP-v0.4.0 pipeline described previously[76].

**Proximity ligation assay.** Proximity ligation assay was performed using Duolink In Situ Orange Starter Kit (Sigma-Aldrich, DUO92102) according to the manufacturer's instructions. Primary antibodies, including anti-p53 antibody (Santa Cruz, sc-126), anti-SVIL antibody (Sigma-Aldrich, S8695), and anti-MLL1 antibody (Novus Biologicals, NBP2-55237), were incubated overnight at 4 °C. Images were taken under a Leica DMi8 microscope.

**m⁶A methylated RNA immunoprecipitation sequencing (m⁶A MeRIP-Seq).** m⁶A MeRIP-Seq was performed using the EpiMark N6-Methyladenosine Enrichment Kit (NEB, #E1610) according to the manufacturer's instructions. Sequencing was performed on an Illumina NextSeq 550 platform at the UTHealth Houston Cancer Genomics Center.

**Xenotransplantation experiments.** The animal experiments were approved by the Animal Welfare Committee, The University of Texas Health Science Center at Houston, Houston, TX, USA (AWC-20-0053). Orthotopic transplantation of iPSC-derived organoids was performed using a method described for brain organoid transplantation into 8-week-old male Foxn1[nu] mice[77]. Mice were induced into anesthesia with 5% isoflurane and maintained with 1.5% isoflurane in oxygen. An approximately 1 mm² craniotomy above the left cerebral cortex at the intersection between the sagittal and lambdoid sutures was performed using a micromotor drill. After removing the meninges, the underlying brain tissue was aspirated using a 23 G blunt needle to create a 1-mm³ cavity. A single organoid was transferred into the cavity by an orifice

pipette tip and sealed by Fibrin. Each transplanted organoid was of similar size at ~1.5 mm in diameter. Animals were weighed twice a week and euthanized immediately after weight loss and/or the onset of neurological symptoms. The perfusion surgery was performed at 60 days after organoid transplantation.

Prior to the perfusion surgery, a ketamine/xylazine mixture (up to 80 mg/kg body weight ketamine and 10 mg/kg body weight xylazine) was administered via intraperitoneal injection. Mice were transcardially perfused with 8 ml cold 0.1 M phosphate buffer followed by 8 ml cold 4% formaldehyde in 0.1 M phosphate buffer (pH 7.4). Brains were carefully removed from the skull and fixed in 4% formaldehyde at 4 °C overnight, washed with DPBS, and cryoprotected in 30% sucrose (w/v) for 48 h at 4 °C. Brains were placed in plastic cryomolds and snap-frozen in Optimal Cutting Temperature (OCT) compound (VWR, 25608-930) on dry ice. Frozen brains were stored at −80 °C until processing.

**Tissue processing and immunostaining.** Serial tissue sections (16 μm for organoids and 35 μm for xenografted rodent brains) were sliced using a cryostat. The tissue sections were washed with TBS containing 0.1% Tween 20 (v/v). Tissue sections were permeabilized and non-specific binding was blocked using a TBS solution containing 10% donkey serum (v/v), 0.5% Triton X-100 (v/v), 3% mouse-on-mouse blocking reagent (v/v) for 1 h at room temperature. The tissue sections were incubated with primary antibodies diluted in TBST with 10% donkey serum (v/v) and 0.1% Triton X-100 (v/v) overnight at 4 °C. After washing in TBST, the tissue sections were incubated with secondary antibodies in TBST with 10% donkey serum (v/v) and 0.1% Triton X-100 (v/v) for 1.5 h at room temperature. After washing with TBST, sections were incubated with TrueBlack reagent (Biotium) diluted 1:20 in 70% ethanol for 3 min to block autofluorescence. After washing with PBS, slides were mounted in a mounting solution (Vector Laboratories), coverslipped, and sealed with nail polish.

**Tumor xenograft assay**

The tumor xenograft assay was performed using a method described previously[26,46]. LNZ308-p53(G245D) glioma lines were transduced with indicated shRNAs or genes, and inoculated into 8-week-old female *Foxn1^nu* mice subcutaneously. Tumor size was measured in two dimensions using a Vernier caliper, and tumor volume was calculated as: $\frac{1}{2} \times \text{Length} \times \text{Width}^2$. As per the guidelines set by the UTHealth Houston Animal Welfare Committee, tumors were allowed to measure up to 20 mm in each dimension. Upon reaching the final tumor measurement time point, the animal will be humanely euthanized using carbon dioxide, followed by cervical dislocation to ensure swift and painless passing.

**Enrichr analysis.** GO biological process, GO molecular function, and KEGG pathway analysis for Fig. 6h and Supplementary Fig. 4d, 7b, c and Epigenomics Roadmap HM ChIP-seq analysis for Fig. 4a were performed using Enrichr (https://maayanlab.cloud/Enrichr/)[78].

**TCGA analysis.** The HiSeqV2 expression datasets and clinical data of LGG, GBM, BRCA, and READ cancers from TCGA were downloaded from UCSC Xena and used to test the expression profiles of *p21*, *PUMA*, and *YTHDF2* between samples with p53 WT and mutation groups. Wilcoxon rank sum test was applied to compare the differential expression of genes between the two groups. Survival analyses were performed using Data Visualization Tools for Brain Tumor Database (GlioVis) (http://gliovis.bioinfo.cnio.es/)[79] using the Gravendeel, TCGA GBM/LGG, and Rembrandt datasets. To examine the YTHDF2-downstream targets associated with survival and hazard ratio, each of the 41 genes was scaled to a z-score across all samples. The average values of the gene z-scores for each sample represented the gene signature score for the sample. The samples were sorted based on

gene signature scores, classified into two groups (high and low expression groups) by the median value, and survival analysis was conducted for clinical survival data based on previously described methods[6]. The hazard ratio (HR) values were recorded and plotted together with 95% CI lower and upper boundaries.

**Clinical samples and IHC staining.** All human brain tumor tissue samples were acquired from the Taipei Veterans General Hospital after obtaining patient informed consent. All procedures involving tissue acquisition were conducted following the tenets of the Declaration of Helsinki and approval by the Institutional Ethics Committee/Institutional Review Board of Taipei Veterans General Hospital. Immunohistochemical staining was performed following the protocol described previously[80]. In brief, tissue sections were incubated with antibodies against YTHDF2 overnight at 4 °C after antigen retrieval and blocking. Human brain tumor specimens were incubated with antibodies to YTHDF2 and then incubated with HRP-conjugated secondary antibodies. The detection of hydrogen peroxide by DAB staining was performed as a chromogenic detection. Differences in YTHDF2 among brain tumor sections were analyzed by Chi-square test.

**Processing of mRNA samples for mass spectrometry.** The preparation of mRNA samples for LC-MS/MS analysis was described previously[81]. Briefly, total RNA was isolated using TRIzol reagent following the manufacturer's instructions. Poly(A) mRNA was further isolated using a magnetic mRNA isolation kit (NEB, S1550S) following the manufacturer's instructions. In total, 500 ng of mRNA was processed by de-capping with 0.5U of Cap-Clip enzyme (Cellscript) at 37 °C for 1 h, digesting with 0.5U of nuclease P1 (Sigma-Aldrich) at 37 °C for 2 h, and then dephosphorylating with 1U of shrimp alkaline phosphatase (rSAP; NEB, M0371S) at 37 °C for 1 h.

**LC-MS/MS for determination of the m⁶A/A ratio.** Standards of m⁶A and adenosine (A) were purchased from MedChemExpress and Sigma-Aldrich, respectively. The standards were dissolved in 30% methanol and serial dilution was carried out with methanol to acquire working standard solutions for the required concentrations. The multiple reaction monitoring (MRM) transitions of m⁶A and adenosine were carried out by a Waters XEVO TQ mass spectrometer with an ESI ionization in positive electrospray ionization (ESI+) mode. The flow rate was set at 10 μl/min. The infusion analysis resulted in MRM transitions of m⁶A being $m/z$ 282 to $m/z$ 150 and that of A being $m/z$ 268 to $m/z$ 136.

The LC-MS-MS system consists of an ACQUITY H-Class UPLC system (Waters, Milford, MA, USA) coupled with a XEVO TQ mass spectrometer with an ESI ionization source (Waters, Milford, MA, USA). The nucleoside samples were thawed and loaded into vials for triplicate analysis. The injection volume was 1 μl. Samples were then separated through the UPLC and a Waters Atlantis C18 column (5 μm, 2.1 mm × 150 mm) with a flow rate of 0.3 mL/min. The oven temperature of the column was 35 °C. A binary gradient system consisting of mobile phases A and B contained 0.1% formic acid (FA) in deionized water and 0.1% FA in methanol, respectively. The isocratic was programmed as follows: 0–3 min, 30% phase B. The mass spectrometer was operated under positive ionization using MRM mode. The MRM transitions were monitored as follows: $m/z$ 282 to $m/z$ 150 for m⁶A (cone voltage: 24 V; collision energy (CE): 20 V; dwell time: 0.025 s) and $m/z$ 268 to $m/z$ 136 for A (cone voltage: 26 V; collision energy (CE): 22 V; dwell time: 0.025 s). The following MS parameters were used: capillary voltage: 3.5 kV; desolvation temperature: 350 °C; desolvation gas flow: 600 L/h; cone gas flow: 50 L/h; collision gas flow: 0.25 mL/min. Both Q1 and Q3 quadrupoles were maintained at quantitative resolution. MassLynx V4.1 software was used for peak areas quantification and data processing.

**Statistical analysis and reproducibility.** Statistical details including values of *n*, statistical tests, significance definitions, and dispersion measures of experiments were included in the figure legends. The data are presented as the mean ± SEM. The experiments were performed at least three times. No statistical methods were used to predetermine the sample size. Student's *t* test, Multiple *t* test, one-way ANOVA with Tukey's multiple comparison test, and two-way ANOVA with Bonferroni's multiple comparison test were applied to determine the statistical significance of the experiments. The significance (*P* value) was calculated using the stated tests in the figure legends. Excel and GraphPad Prism 8.0 were used for statistical analysis.

### Reporting summary
Further information on research design is available in the Nature Portfolio Reporting Summary linked to this article.

## Data availability
The data supporting the findings of this study are available within the article and its Supplementary Information. The RNA-seq, ChIP-seq, m⁶A MeRIP-Seq, and YTHDF2 eCLIP-seq data generated in this study have been deposited in the GEO repository under accession number GSE163088. Source data are provided with this paper.

## Code availability
The scripts for eCLIP analysis are available at https://github.com/VanNostrandLab/eclip. The scripts for TCGA analysis are available at https://github.com/huruifeng/m6A_p53_TCGA.

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

## Acknowledgements

We thank Drs. Carol Prives, Markus T. Bohnsack, Elizabeth J. Luna, and Zhiyou Fang for plasmids and Dr. Jiaofang Shao and Lee laboratory members for technical assistance and/or discussion. We acknowledge the Center for Advanced Microscopy and Cancer Genomics Center (supported by CPRIT RP180734) at the University of Texas Health

Science Center at Houston and the Optical Imaging and Vital Microscopy Core at Baylor College of Medicine for their support with imaging and data processing, and The University of Texas MD Anderson Proteomics and Metabolomics Facility (supported by MD Anderson Cancer Center NIH High-End Instrumentation program grant 1S10OD012304-01 and CPRIT Core Facility Grant RP130397) for the mass spectrometry analysis. This work was mainly supported by UTHealth start-up funds (D.-F.L. (37516-11998) and R.Z. (37516-11999)) and Cancer Prevention and Research Institute of Texas (CPRIT) RR16009 (D.-F.L.). A.X. was a CPRIT Postdoctoral Fellow in the Biomedical Informatics, Genomics and Translational Cancer Research Training Program (BIG-TCR, CPRIT grant RP210045). C.D.H. and Y.Y. were supported by the Andrew Sabin Family Foundation Fellowship. E.L.V.N. was supported by the CPRIT (RR200040). D.-F.L. was supported by the CPRIT (RR160019), NIH/NCI R01CA246130, Rolanette and Berdon Lawrence bone disease program of Texas, and Pablove Foundation childhood cancer research grant (690785). D.Z. was supported by DoD Horizon Award (W81XWH-20-1-0389). Z.Z. was partially supported by NIH/NLM (R01LM012806) and CPRIT (RP180734 and RP210045). W.L., E.L.V.N., and D.-F.L. are CPRIT Scholars in Cancer Research.

## Author contributions

A.X., M.L., M.-F.H., Y.L., D.Z., T.R., and R. Z. conducted all experiments. A.X., M.L., J.A.G., M.-C.H., Z.Z., R.Z., and D.-F.L. conceived and designed the study and interpreted the results. A.X., M.L. M.-F.H., Y.Z., F.A., N.C.B., and D.-F.L. performed RNA-seq and m$^6$A MeRIP-seq analysis. A.X., M.L., F.X., W. L., and N.C.B. conducted ChIP-seq analysis. Z.L., F.Y., and E.L.V.N. conducted eCLIP assays. C.-S.C., Y.-P.Y., and S.-H.C. performed the IHC assay. C.-W.H. conducted the light sheet image analysis. J.T., Y.Y., and C.D.H. conducted mutation analysis. W.-C.W. and C.-C.L. conducted mass spectrometry analysis. R.H., P.J., and Z.Z. conducted TCGA analysis., Y.-W.C., and J.-J.Z. provided critical reagents. M.L., A.X., J.A.G., D.A.B., R.Z., and D.-F.L. wrote the manuscript with critical suggestions from all authors.

## Competing interests

E.L.V.N. is a co-founder, member of the Board of Directors, on the SAB, equity holder, and paid consultant for Eclipse BioInnovations. E.L.V.N.'s interests have been reviewed and approved by the Baylor College of Medicine in accordance with its conflict of interest policies. The remaining authors declare no competing interests.

## Additional information

[1]Department of Integrative Biology and Pharmacology, McGovern Medical School, The University of Texas Health Science Center at Houston, Houston, TX 77030, USA. [2]The University of Texas MD Anderson Cancer Center UTHealth Houston Graduate School of Biomedical Sciences, Houston, TX 77030, USA. [3]College of Science, Harbin Institute of Technology (Shenzhen), Shenzhen, Guangdong 518055, China. [4]Center for Precision Health, School of Biomedical Informatics, The University of Texas Health Science Center at Houston, Houston, TX 77030, USA. [5]Department of Obstetrics & Gynecology and Women's Health, Einstein/Montefiore Medical Center, Bronx, NY 10461, USA. [6]Department of Medical Research, Taipei Veterans General Hospital, Taipei 112, Taiwan. [7]College of Medicine, National Yang Ming Chiao Tung University, Taipei 112, Taiwan. [8]Institute of Molecular Biology, National Chung Hsing University, Taichung 40227, Taiwan. [9]Verna & Marrs McLean Department of Biochemistry & Molecular Biology and Therapeutic Innovation Center, Baylor College of Medicine, Houston, TX 77030, USA. [10]Department of Molecular Physiology and Biophysics, Baylor College of Medicine, Houston, TX 77030, USA. [11]Department of Epidemiology, The University of Texas MD Anderson Cancer Center, Houston, TX 77030, USA. [12]Department of Biochemistry and Molecular Biology, McGovern Medical School, The University of Texas Health Science Center at Houston, Houston, TX 77030, USA. [13]Department of Neurology, Renaissance School of Medicine at Stony Brook University, Stony Brook, NY 11794, USA. [14]Department of Otolaryngology, Icahn School of Medicine at Mount Sinai, New York, NY 10029, USA. [15]Department of Cell, Developmental and Regenerative Biology, Icahn School of Medicine at Mount Sinai, New York, NY 10029, USA. [16]Black Family Stem Cell Institute, Icahn School of Medicine at Mount Sinai, New York, NY 10029, USA. [17]Institute for Airway Sciences, Icahn School of Medicine at Mount Sinai, New York, NY 10029, USA. [18]Department of Neurosurgery, McGovern Medical School, The University of Texas Health Science Center at Houston, Houston, TX 77030, USA. [19]Wallenberg Centre for Molecular Medicine (WCMM), Umea University, SE-901 85 Umea, Sweden. [20]Department of Molecular Biology, Umea University, SE-901 85 Umea, Sweden. [21]Neural Stem Cell Institute, Rensselaer, NY 12144, USA. [22]Graduate institute of Chinese Medical Science, China Medical University, Taichung 40402, Taiwan. [23]Ph.D. Program in Translational Medicine and Rong Hsing Research Center for Translational Medicine, National Chung Hsing University, Taichung 40227, Taiwan. [24]Graduate Institute of Biomedical Sciences and Center for Molecular Medicine, and Office of the President, China Medical University, Taichung 404, Taiwan. [25]Department of Biotechnology, Asia University, Taichung 413, Taiwan. [26]Center for Stem Cell and Regenerative Medicine, The Brown Foundation Institute of Molecular Medicine for the Prevention of Human Diseases, The University of Texas Health Science Center at Houston, Houston, TX 77030, USA. [27]These authors contributed equally: An Xu, Mo Liu. ✉e-mail: ruiying.zhao@uth.tmc.edu; dung-fang.lee@uth.tmc.edu

