## [Peer Review File · Nature Communications]

Rewired m⁶A epitranscriptomic networks link mutant p53 to neoplastic transformationEditorial Note: This manuscript has been previously reviewed at another journal that is not operating a transparent peer review scheme. This document only contains reviewer comments and rebuttal letters for versions considered at *Nature Communications*.

REVIEWER COMMENTS

Reviewer #1 (Remarks to the Author):

The authors have addressed most of the concerns and comments, and the addition of new experiments has significantly improved the manuscript. I only have two remaining concerns;

1. the authors show that high YTHDF2 expression correlates with poor prognosis in wtp53 glioma patients and not mutp53 patients - which questions the relevance of mutp53 in driving YTHDF2-mediated oncogenic functions in vivo, and that this mechanism may not be the contributing factor for mutp53 GOF.

2. the lack of tumour formation in vivo using their model - however, you could claim that hyperplasia is a prelude to transformation but it is not an absolute proof - perhaps considering modifying the title of the manuscript?

Reviewer #2 (Remarks to the Author):

The authors have conducted additional experiments and data analysis, and have thoroughly addressed our concerns. This is an interesting and innovative study. Thus, I would like to recommend the publication of this paper in Nature Communications.

Reviewer #3 (Remarks to the Author):

The authors have addressed many of my concerns, and new results support the conclusions. However, the quality of some data should be further improved before publication. In particular:

- To visualize the mutant p53/SVIL/MLL1 protein complex, the authors performed PLA in LFS astrocytes (Fig. 4d). They should indicate: i) what the control is, ii) assess the effect of knockdown of mutp53, of SVIL, and of MLL1. They also performed coIP to demonstrate the SVIL-dependent formation of p53/SVIL/MLL1 complex. This experiment should include IgG and input controls

- To demonstrate YTHDF2-dependent regulation of CDKN2B and SPOCK2, the authors performed immunofluorescence staining in LFS cerebral organoids engrafted in the mouse cortex (Fig. 5j and Extended Data Fig. 6c). TO improve the quality of these data, I recommend to include, as controls, YTHDF2 staining and markers to distinguish the engrafted tissue.

- The annotation and legends of some figures should be improved to improve readability. For example, in Fig3, please explain the meaning of "peak" and "up" annotations.

Point-by-point response to reviewers' and editorial comments (Ms. No. NCOMMS-22-40872-T)

We thank all the reviewers for their positive assessment of our work and their highly constructive comments. This has helped us design new experiments and revise the manuscript to greatly improve the clarity and the impact of our findings on the field. Based on the reviewers' comments, we performed new experiments, including **4** new figures (Fig. 4e, f and Supplementary Fig. 6c, d) and **3** revised figures (Fig. 3c, d, 4d, and 5j), and revised the manuscript accordingly. We trust that all comments have been addressed satisfactorily either through additional experiments or through better explanations to clarify specific points.

Reviewers' Comments:

Reviewer #1:

1. *The authors show that high YTHDF2 expression correlates with poor prognosis in wtp53 glioma patients and not mutp53 patients - which questions the relevance of mutp53 in driving YTHDF2-mediated oncogenic functions in vivo, and that this mechanism may not be the contributing factor for mutp53 GOF.*

Response: Thank you for this comment. Our results indeed demonstrate that YTHDF2 functions as an oncoprotein in glioma development and its expression is correlated to poor prognosis (Fig. 2f). Mutant p53 regulates the YTHDF2-associated m⁶A epitranscriptome, which contributes to glioma initiation and progression. In p53 mutant glioma specimens, YTHDF2 expression is at a relatively high level due to mutant p53-mediated YTHDF2 transcription upregulation (Fig 7b and Supplementary Fig. 8a, TCGA glioma data). Therefore, the additional changes of YTHDF2 expression (moderately high vs high) in p53 mutant glioma patients only have limited effects on patient survival since elevated YTHDF2 by mutant p53 is a sufficient dominant influencer of patient survival among p53 mutant glioma patients. To make it clear, we revised our conclusion to "Interestingly, high expression of YTHDF2 was associated with poor prognosis in glioma patients with wild-type p53 but not mutant p53 (Fig. 2g), suggesting that high YTHDF2 expression is correlated to poor prognosis in p53 wild-type glioma patients and the elevated level of YTHDF2 by mutant p53 is a sufficient dominant influencer of patient survival among p53 mutant glioma patients." (lines 239-241 of the revised manuscript)

2. *The lack of tumour formation in vivo using their model - however, you could claim that hyperplasia is a prelude to transformation but it is not an absolute proof - perhaps considering modifying the title of the manuscript?*

Response: We sincerely appreciate this comment on the title. We have demonstrated that LFS iPSC-derived astrocytes possess anchorage-independent growth ability (Fig. 2b) by soft agar colony formation assay, a well-recognized method to evaluate neoplastic transformation. Therefore, we believe the current title "Rewired m⁶A epitranscriptomic networks link mutant p53 to neoplastic transformation" fits our conclusion.

Reviewer #3:

1. *To visualize the mutant p53/SVIL/MLL1 protein complex, the authors performed PLA in LFS astrocytes (Fig. 4d). They should indicate: i) what the control is, ii) assess the effect of knockdown of mutp53, of SVIL, and of MLL1. They also performed colIP to demonstrate the SVIL-dependent formation of p53/SVIL/MLL1 complex. This experiment should include IgG and input controls.*

Response: Thank you for these comments. **First**, the control is the primary antibody IgG. We have relabeled it in Fig. 4d and explained it in the figure legend. **Second**, following the reviewer's

suggestion, we have assessed the effect of knockdown of mutant p53, SVIL, or MLL1 on mutant p53/SVIL/MLL1 protein complex by PLA assay. Our results indicate that depletion of p53 or SVIL hampers p53(G245D)/SVIL/MLL1 complex formation but that knockdown of MLL1 does not interfere with the interaction between p53(G245D) and SVIL (Fig. 4f). **Third**, we also include IgG and input controls in our co-IP study (Fig. 4e)

2. *To demonstrate YTHDF2-dependent regulation of CDKN2B and SPOCK2, the authors performed immunofluorescence staining in LFS cerebral organoids engrafted in the mouse cortex (Fig. 5j and Extended Data Fig. 6c). To improve the quality of these data, I recommend to include, as controls, YTHDF2 staining and markers to distinguish the engrafted tissue.*

Response: Following the reviewer's suggestion, we have performed YTHDF2 and SPOCK2 staining by co-staining YTHDF2 (rabbit IgG) or SPOCK2 (rabbit IgG) with hNuclei (mouse IgG). The CDKN2B staining uses antibodies against human species (mouse monoclonal IgG (DCS114.1), Thermo Fisher Scientific, MA1-12294); thus, it does not need to co-stain with human markers (both hNuclei and STEM121 antibodies are mouse monoclonal antibodies) to distinguish the engrafted tissue. The results are included in Fig. 5j and Supplementary Fig. 6c, d.

3. *The annotation and legends of some figures should be improved to improve readability. For example, in Fig3, please explain the meaning of "peak" and "up" annotations.*

Response: Thank you for this comment. We have gone through the annotation and legends of figures to improve readability. For example, the legends of Fig 3c and d are revised as "**c** ChIP-qPCR validation of p53 and p53(G245D) binding peaks at the identified *YTHDF2* promoter. (Upper panel) Schematic of amplicon locations of *p21* and *YTHDF2* genes. p53/mutant p53 binding peak (blue) and upstream non-p53/mutant p53 binding regions (green) used for ChIP-qPCR validation. (Lower panel) ChIP-qPCR at *YTHDF2* peak sites and upstream non-p53/mutant p53 binding regions confirm specific enrichment of *YTHDF2* among p53 peak regions in both WT and LFS astrocytes, but reduced enrichment of *p21* at peak regions in LFS compared to WT astrocytes (n=3). **d.** ChIP-qPCR indicates that p53 and various p53 mutants bind to the *YTHDF2* promoter but not the upstream non-p53/mutant p53 binding region (n=3)." (lines 1209-1216 of the revised manuscript)

REVIEWERS' COMMENTS

Reviewer #1 (Remarks to the Author):

The authors have addressed my concerns and comments satisfactory

Reviewer #3 (Remarks to the Author):

The authors have addressed the concerns. I have one comment only related to the quality of the images in Fig.4f. that should be improved.

Reviewers' Comments:

Reviewer #3:

1. *The authors have addressed the concerns. I have one comment only related to the quality of the images in Fig.4f. that should be improved.*

Response: Thank you for this comment. We have improved the resolution of the PLA staining images **Fig. 4d** and **4f**).